# The neurohormone tyramine stimulates the secretion of an insulin-like peptide from the *Caenorhabditis elegans* intestine to modulate the systemic stress response

**Tania Veuthey**[1,2], **Jeremy T. Florman**[3], **Sebastián Giunti**[1,2], **Stefano Romussi**[1,2], **María José De Rosa**[1,2], **Mark J. Alkema**[3]*, **Diego Rayes**[1,2]*

**1** Instituto de Investigaciones Bioquímicas de Bahía Blanca (INIBIBB) CCT UNS-CONICET, Bahía Blanca, Argentina, **2** Departamento de Biología, Bioquímica y Farmacia, Universidad Nacional Del Sur (UNS), Bahía Blanca, Argentina, **3** Department of Neurobiology, University of Massachusetts Chan Medical School, Worcester, Massachusetts, United States of America

* mark.alkema@umassmed.edu (MJA); drayes@criba.edu.ar (DR)

## Abstract

The DAF-2/insulin/insulin-like growth factor signaling (IIS) pathway plays an evolutionarily conserved role in regulating reproductive development, life span, and stress resistance. In *Caenorhabditis elegans*, DAF-2/IIS signaling is modulated by an extensive array of insulin-like peptides (ILPs) with diverse spatial and temporal expression patterns. However, the release dynamics and specific functions of these ILPs in adapting to different environmental conditions remain poorly understood. Here, we show that the ILP, insulin-3 (INS-3), plays a crucial role in modulating the response to various environmental stressors in *C. elegans*. *ins-3* mutants display increased resistance to heat, oxidative stress, and starvation; however, this advantage is countered by slower reproductive development under favorable conditions. We find that *ins-3* expression is downregulated in response to environmental stressors, whereas, the neurohormone tyramine, which is released during the acute flight response, increases *ins-3* expression. We show that tyramine induces intestinal calcium ($Ca^{2+}$) transients through the activation of the TYRA-3 receptor. Our data support a model in which tyramine negatively impacts environmental stress resistance by stimulating the release of INS-3 from the intestine via the activation of a TYRA-3-$G_{\alpha q}$-IP3 pathway. The release of INS-3 systemically activates the DAF-2 pathway, resulting in the inhibition of cytoprotective mechanisms mediated by DAF-16/FOXO. These studies offer mechanistic insights into a brain–gut communication pathway that weighs adaptive strategies to respond to acute and long-term stressors.

## Introduction

In nature, animals face changing environmental conditions that require adaptive responses to successfully survive and reproduce. These adaptive responses are orchestrated by sophisticated regulatory mechanisms that control a spectrum of physiological adaptations [1,2]. For example, when conditions are optimal, i.e., abundant food resources and favorable temperatures, animals

**Data availability statement:** All the raw data and code are available on the Open Science Framework platform (https://osf.io/wfgvs/).

**Funding:** This work was supported by Grants from: -Consejo Nacional de Investigaciones Científicas y Técnicas, Argentina (PIP No. 11220200101606CO) to DR. -Consejo Nacional de Investigaciones Científicas y Técnicas, Argentina (PIBAA 28720210101169CO) to TV. -NIH R01GM140480 to MJA. -Agencia Nacional de Promoción de la Ciencia y la Tecnología ANPCYT Argentina (PICT 2019-0480) to DR. -Agencia Nacional de Promoción de la Ciencia y la Tecnología ANPCYT Argentina (PICT-2021-I-A-00052) to DR. -Agencia Nacional de Promoción de la Ciencia y la Tecnología ANPCYT Argentina (PICT-2017-0566) to MJDR. -Agencia Nacional de Promoción de la Ciencia y la Tecnología ANPCYT Argentina (PICT-2020-1734) to MJDR. -Agencia Nacional de Promoción de la Ciencia y la Tecnología ANPCYT Argentina (PICT 2018-03164) to TV. -Universidad Nacional Del Sur (PGI: 24/B291) to DR. -Universidad Nacional Del Sur (PGI: 24/B344) to TV. -Universidad Nacional Del Sur (PGI: 24/B261) to MJDR. The funders had no role in the study design, data collection, and analysis, decision to publish, or preparation of the manuscript.

**Competing interests:** The authors have declared that no competing interests exist.

**Abbreviations:** cDNA, complementary DNA; CGC, Caenorhabditis Genetics Center; DAF, Abnormal DAuer Formation; DAG, diacylglycerol; DMP, defecation motor program; FOXO, FOrkhead boX O; GFP, green fluorescence protein; HSF-1, Heat Shock Factor-1; IIS, insulin/insulin-like growth factor signaling; ILPs, insulin-like peptides; INS-3, insulin-3; IP3, inositol trisphosphate; ITR-1, inositol trisphosphate receptor 1; L4, fourth larval stage; NGM, Nematode Growth Medium; RICAi, ring interneuron CUV1; RIM, ring interneuron M; RNAi, RNA interference; ROI, region of interest; RT-qPCR, Reverse transcription-quantitative polymerase chain reaction; TDC-1, tyrosine decarboxylase-1; TYRA-3, TYRAmine receptor 3; UV1, uterine-vulval cell 1.

prioritize and accelerate their growth and reproductive development to gain a competitive advantage [3,4]. Conversely, when faced with adverse environmental conditions, such as food deprivation or oxidative stress, animals adopt a different strategy. They reduce their growth and reproductive rates and redirect their energy toward cytoprotection [5,6]. The evolutionarily conserved insulin/insulin-like growth factor signaling (IIS) pathway plays a pivotal role in this strategic plasticity of animals [7–11]. Elevated levels of insulin and insulin-like peptides (ILPs) are associated with cell growth and development [7,12]. In contrast, reduced insulin levels are associated with increased life span and greater resistance to prolonged environmental stressors such as high temperature or oxidation [13–16]. The mechanisms by which the release of ILPs is coordinated in response to environmental conditions remain largely unknown in all animals.

The complexity of mammalian stress physiology makes it challenging to elucidate the fundamental mechanisms underlying the systemic control of cellular defense processes. Genetic studies in the nematode *Caenorhabditis elegans* have been instrumental in revealing the importance of the conserved IIS pathway in controlling both life span and stress resistance [15,17,18]. Although *C. elegans* has a single insulin receptor, DAF-2, its genome encodes 40 ILPs that can interact with this receptor [19]. Downregulation of DAF-2 typically extends life span and increases the resistance to environmental stressors [17]. ILPs have been reported to modulate aversive olfactory learning [20], life span, dauer formation [19,21,22], and germ cell proliferation through the IIS pathway [23]. However, mutation or deletion of single insulin genes failed to completely recapitulate the phenotypes observed in *daf-2* mutants [24]. Furthermore, the overexpression of each of the 40 insulin genes revealed that these ILPs can act not only as strong agonists, but also as weak agonists, antagonists, or even exhibit pleiotropic functions on the DAF-2/IIS pathway [25]. Therefore, deciphering the specific roles of individual ILPs is challenging. As in most animals, where ILPs act as hormones, in *C. elegans*, ILPs are secreted into the pseudocoelom, the fluid-filled body cavity, where they act as long-range signaling molecules [7]. Importantly, the mechanisms underlying ILP secretion in *C. elegans* are similar to those in mammals, and conserved factors such as ASNA-1, HID-1, and GON-1 have been shown to play a role in insulin secretion [26–28].

Under acute life-threatening situations, such as imminent predation or aggression, organisms initiate an energy-demanding, short-lived "flight response" to increase the chances of escape [29–31]. We have previously uncovered a novel brain–gut communication pathway in *C. elegans* in which neural stress hormones released during the flight response negatively impact health by activating the DAF-2/IIS pathway [32]. The flight response triggers the activation of a single pair of neurons, called ring interneuron M (RIM), that release tyramine [32,33]. Tyramine, the invertebrate analog of adrenaline, coordinates different motor programs of the flight response, allowing the worm to escape from predation [34–37]. However, the downregulation of tyramine signaling is important for coping with environmental stressors when behavioral responses alone fail to alleviate the exposure to a threatening stimulus [32]. Specifically, we found that tyramine activates an adrenergic-like receptor, TYRA-3, in the intestine. TYRA-3 activation subsequently leads to the stimulation of the DAF-2/IIS pathway throughout the body. Systemic activation of the DAF-2/IIS pathway may increase the metabolic rate required for the physical and energy demands of the flight response, but it comes at a cost: it inhibits cytoprotective mechanisms needed to cope with longer-lasting environmental stressors. Thus, tyramine acts as a state-dependent neural signal that controls the switch between acute and long-term stress responses. How tyraminergic activation of TYRA-3 in the intestine leads to the systemic stimulation of the DAF-2 pathway is unclear.

Here, we find that activation of the tyraminergic receptor TYRA-3 increases calcium transients in the intestine and induces the release of the ILP insulin-3 (INS-3). We show that *ins-3* mutants are more resistant to environmental stressors such as oxidation, starvation, and heat.

In contrast, during the activation of the flight response, the tyramine-induced release of intestinal INS-3 leads the systemic activation of the DAF-2/IIS pathway, which in turn suppresses cytoprotective mechanisms that rely on DAF-16/FOXO. The tyramine-induced activation of the DAF-2/IIS pathway renders animals more vulnerable to environmental stressors. By elucidating the tyramine-mediated secretion of INS-3, we reveal the molecular sequence underlying the detrimental effects of a sustained flight-stress response in *C. elegans*.

## Results

### *ins-3* mutants are resistant to environmental stressors

The *C. elegans* genome encodes an extensive family of ILP genes. Many of these ILPs are expressed in the intestine [24]. Our previous work suggested that ILPs released from the intestine may play a key role in modulating the stress response [32]. To test this hypothesis, we used RNA interference (RNAi) to silence the expression of intestinal ILPs which have been reported to act as strong agonists of the DAF-2/IIS signaling pathway [25]. We examined resistance to the oxidizing agent $FeSO_4$ (15 mM, 1 h) in animals treated with *ins-3*, *ins-4*, *ins-6*, *ins-32* and *daf-28*, RNAi. RNAi silencing of *ins-3* increased resistance to oxidative stress (Fig 1A). Like *ins-3* RNAi silenced animals, null mutants in the *ins-3* gene are significantly more resistant to oxidative stress than control animals (Fig 1B). *ins-3* mutants are also more resistant to heat stress (35 °C, 4 h) and starvation compared to control animals (Fig 1B and S1 Fig) suggesting that INS-3 is a general inhibitor of the environmental stress response.

We found that a *Pins-3::GFP* transcriptional reporter is predominantly expressed in the intestine and a subset of neurons consistent with previous reports (Fig 2A) [24]. Transgenic expression of *ins-3* under the control of its endogenous promoter in an *ins-3* null mutant background

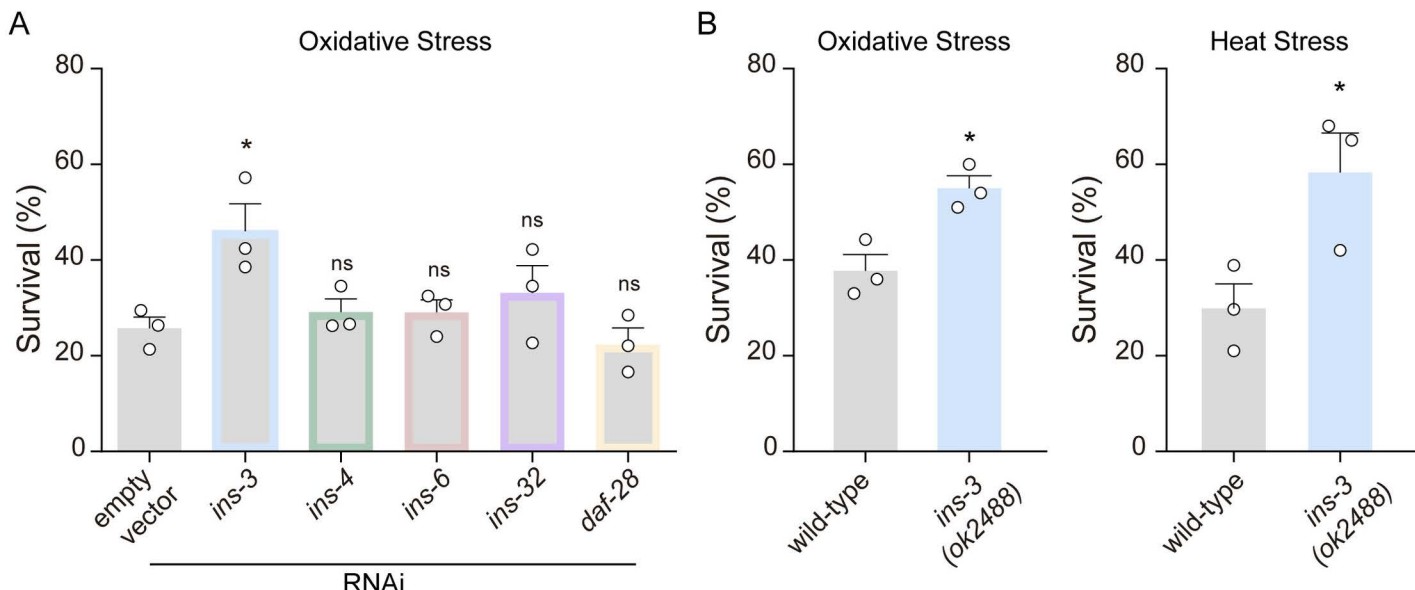

**Fig 1. The deficiency of the insulin-like peptide (ILP) INS-3 confers enhanced resistance to environmental stressors.** (A) Survival percentages to oxidation (15 mM $FeSO_4$, 1 h) in animals subjected to RNAi-mediated silencing of ILPs that are expressed in the intestine, and were previously identified as potent agonists of the DAF-2 receptor. Mean ± s.e.m. Three independent experiments were performed ($n = 3$). Each experiment included 60–80 animals per condition. One-way ANOVA, Holm–Sidak's post hoc test for multiple comparisons vs. empty vector were used. ns, not significant; * $p < 0.05$. RNAi, RNA interference. (B) Survival percentages of wild-type and *ins-3* null mutant worms exposed to oxidation (15 mM $FeSO_4$, 1 h) and heat stress (35 °C, 4 h). A two-tailed Student's *t* test was used. Three independent experiments were performed ($n = 3$). Each experiment included 40–80 animals per condition * $p < 0.05$. The data underlying this figure can be found at https://osf.io/wfgvs/.

**A**

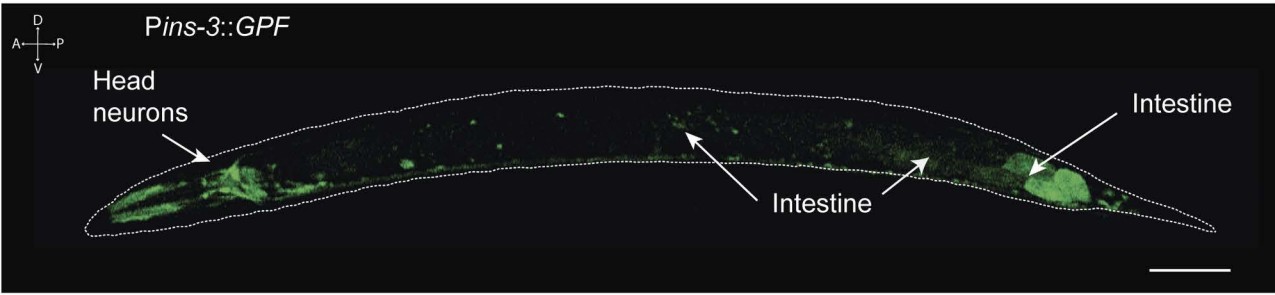

**B**

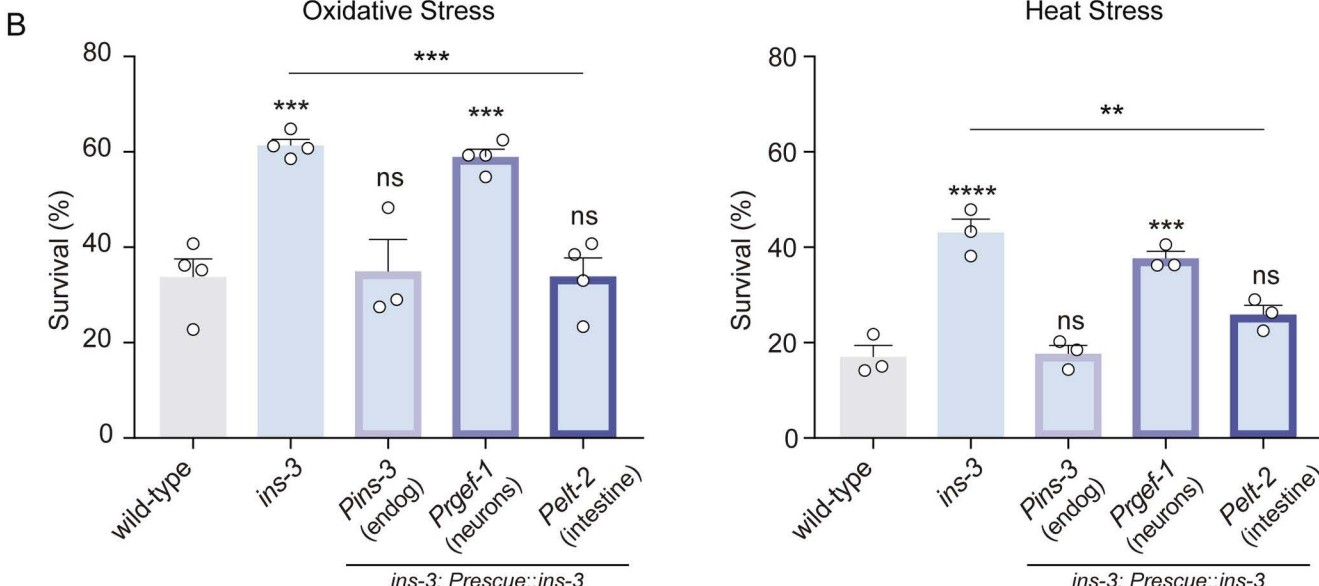

**Fig 2. Intestinal INS-3 modulates the stress response.** (A) Expression of *Pins-3::GFP* reporter in an L4 worm. Scale bar 50 µm. **(B)** Stress resistance of *ins-3* mutant animals expressing *ins-3* cDNA driven by *Pins-3* (endogenous), *Prgef-1* (pan-neuronal) or *Pelt-2* (intestinal) promoters upon exposure to oxidative stress (left) or heat stress (right). Mean ± s.e.m. Three to four independent experiments were performed ($n = 3$−4). Each experiment included 40–80 animals per condition. One-way ANOVA, Holm–Sidak's post hoc test for multiple comparisons vs. wild type were used in both graphs. Two-tailed Student's *t* test was used in both graphs for comparison between *ins-3* null mutant and *ins-3*; P*elt-2*::INS-3. ns: not significant, **$p < 0.01$, ***$p < 0.001$, ****$p < 0.0001$. The data underlying this figure can be found at https://osf.io/wfgvs/.

restored oxidative stress sensitivity to wild-type levels (Fig 2B). To determine where *ins-3* acts to modulate the stress response, we performed tissue-specific rescue experiments. Expression of *ins-3* in the intestine but not in neurons, completely restored oxidative and heat stress sensitivity of *ins-3* null mutants to wild-type levels (Fig 2B). These findings indicate that INS-3 release from the intestine inhibits the animal's capacity to cope with environmental stressors.

## INS-3 stimulates the DAF-2/IIS signaling pathway

ILP agonists can activate the DAF-2/IIS receptor, resulting in the phosphorylation of DAF-16 and its retention in the cytosol [13]. To investigate whether intestinal INS-3 systemically activates the DAF-2/IIS pathway, we examined the subcellular localization of the transcription

factor DAF-16/FOXO using the translational *Pdaf-16::DAF-16::GFP* reporter. Under basal conditions, DAF-16 was predominantly localized in the cytoplasm in both wild-type and *ins-3* null mutant backgrounds (S2 Fig). Following mild stress induction (35 °C, 10 m), *ins-3* mutants showed a significantly higher nuclear accumulation of DAF-16 compared to wild-type animals (Fig 3A). This suggests that in *ins-3* mutants, DAF-16 is more readily activated in response to stressors. Notably, the tissue-specific rescue of *ins-3* in the intestine reduced

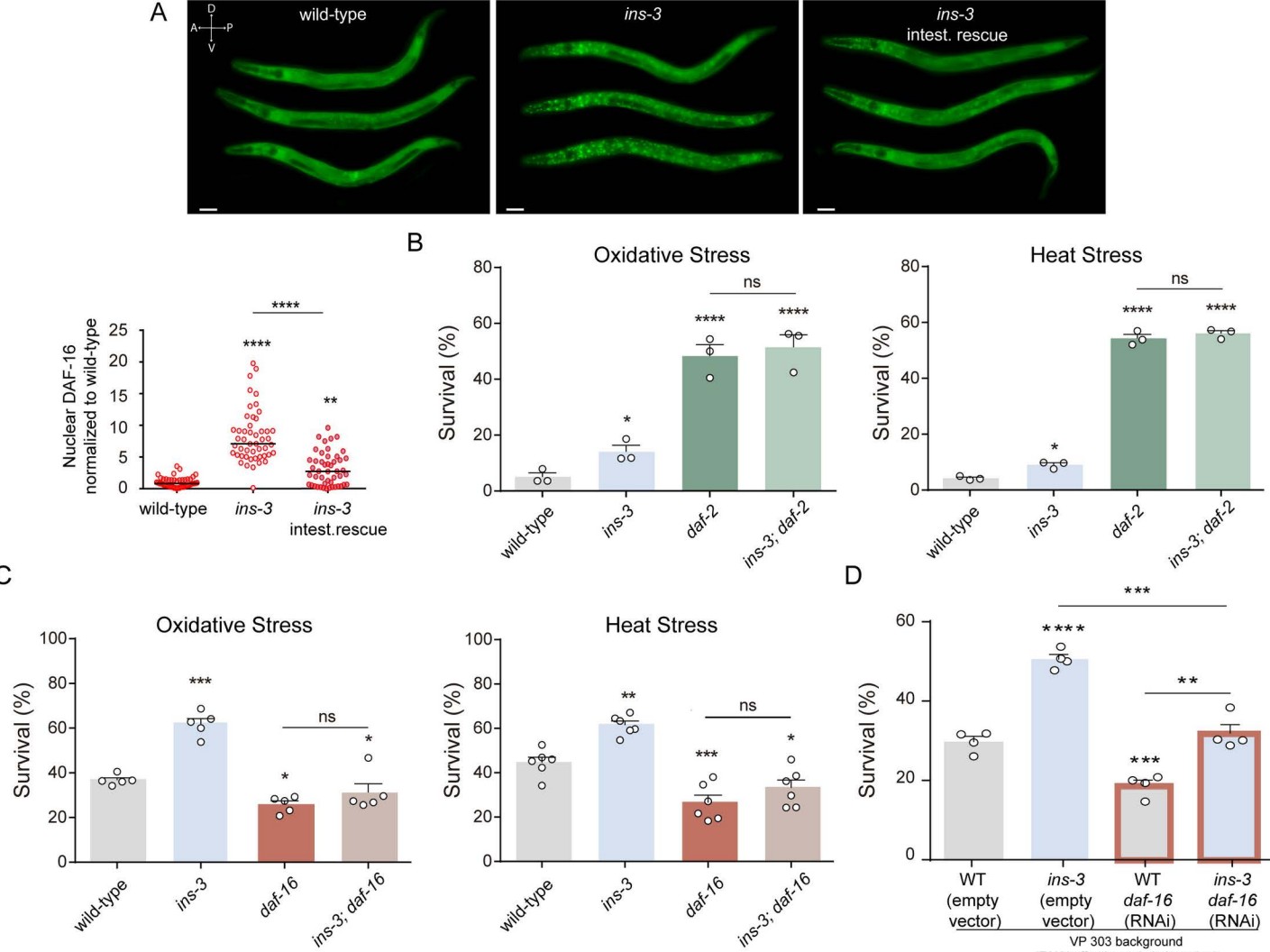

**Fig 3. INS-3 inhibits environmental stress response through the inactivation of DAF-16/FOXO.** (**A**) Up: Representative fluorescence images (20×) depicting the localization of DAF-16a/b::GFP after a short heat exposure (35 °C, 10 min) in wild-type, *ins-3* null mutants, and intestinal rescue of *ins-3* backgrounds. Scale bar, 50 μm. Bottom: Corresponding scatter dot plots with the number of cells with nuclear DAF-16 per animal (normalized to wild-type). Line shows the median. $n = 45$–50 animals per condition distributed across (3–4) independent experiments were evaluated. One-way ANOVA with Dunn's multiple comparisons post hoc test was used. \*\**p* < 0.01, \*\*\*\**p* < 0.0001. (**B**) Survival percentages of wild-type animals, *ins-3* single mutants, *daf-2* single mutants, and *ins-3;daf-2* double mutants exposed to oxidative (left) and heat stress (right). Results are shown as mean ± s.e.m. Three independent experiments were performed ($n = 3$). Each experiment included 65–100 animals per condition. \**p* < 0.05, \*\*\*\**p* < 0.0001. (**C**) Survival percentages of wild-type animals, *ins-3* single mutants, *daf-16* single mutants, and *ins-3;daf-16* double mutants exposed to oxidative (left) and heat stress (right). Results are shown as mean ± s.e.m. Five ($n = 5$) and six ($n = 6$) independent experiments were performed for oxidative stress and heat, respectively. Each experiment included 50–100 animals per condition. \**p* < 0.05, \*\*\**p* < 0.001. (**D**) Oxidation resistance of animals subjected to intestinal RNAi-mediated silencing of *daf-16*. The VP303 (*kbIs7 [nhx-2p::RDE-1 + rol-6(su1006)]*) background allows RNAi silencing only in the intestine [55,109]. Results are shown as mean ± s.e.m. Four independent experiments were performed ($n = 4$). Each experiment included 30–80 worms per condition. \*\**p* < 0.01, \*\*\**p* < 0.001, \*\*\*\**p* < 0.0001. The data underlying this figure can be found at https://osf.io/wfgvs/.

nuclear DAF-16 accumulation (Fig 3A). In addition, we found that *ins-3* mutants produce fewer offspring and exhibits slightly slower development compared to wild-type animals (S3 Fig), consistent with the downregulation of the DAF-2/IIS pathway [38].

To determine whether the effects of *ins-3* on the stress response depend on its ability to activate DAF-2, we analyzed the stress resistance of *ins-3;daf-2* double mutants. Since *daf-2* null mutants are extremely resistant to stress [15,32], we used harsh stress conditions (35 °C, 7 h for heat stress and 15 mM $FeSO_4$, 3 h for oxidative stress) to detect potential additive effects in the double mutants. The absence of *ins-3* does not further enhance the stress resistance of *daf-2* mutants, indicating that INS-3 modulates stress resistance through the activation of DAF-2 (Fig 3B).

In sharp contrast to *daf-2* mutants, *daf-16* mutants exhibit increased sensitivity to heat and oxidative stress compared to wild-type animals [39–42]. *ins-3;daf-16* double mutants displayed stress resistance levels similar to *daf-16* single mutants (Fig 3C), suggesting that the stress resistance observed in *ins-3* mutants depends on DAF-16 activation. Our results support the conclusion that INS-3 activation of DAF-2 inhibits DAF-16 translocation to the nucleus, impairing the resistance to environmental stressors.

Since *ins-3* is required in the intestine to modulate stress resistance, we investigated whether the role of *daf-16* is also restricted to the intestine. We performed specific RNAi-mediated knockdown of *daf-16* in the intestine in both wild-type and *ins-3* null mutant backgrounds. We found that stress resistance in *ins-3* mutants with intestinal *daf-16* knock-down (*ins-3;daf-16 RNAi^{intestine}*) is lower compared to *ins-3* mutants (Fig 3D). Nevertheless, *ins-3;daf-16 RNAi^{intestine}* animals are more stress resistant than wild-type animals with intestine-specific *daf-16* knockdown (*daf-16 RNAi^{intestine}*) (Fig 3D). This suggests that while intestinal activation of DAF-16 plays a key role in the enhanced stress resistance observed in *ins-3* mutants, the systemic activation of DAF-16 in other tissues is required as well.

Activation of DAF-2 affects the expression of many cytoprotective proteins [43]. One such protein is HSP-16.2, which acts as an effector for DAF-16 and Heat Shock Factor-1 (HSF-1) [43–45]. Expression of the transcriptional reporter of *hsp-16.2* (P*hsp-16.2::GFP*) is induced by heat [32,46]. Under basal conditions, *hsp-16.2::GFP* fluorescence intensity was similar between wild-type and *ins-3* mutant animals (S4A Fig). However, after mild heat stress (35 °C, 15 m), *hsp-16.2* expression was markedly increased in *ins-3* mutants compared to the wild-type (S4 Fig). Intestinal rescue of *ins-3* expression decreased *hsp-16.2* expression of *ins-3* mutants, albeit not to wild-type levels (S4 Fig).

Since *hsp-16.2* expression is regulated by both DAF-16 and HSF-1, we performed genetic experiments using the *hsf-1(sy441)* loss-of-function mutant. Animals were exposed to the same oxidative- (15 mM $FeSO_4$, 1 h) and heat-stress (35 °C, 4 h) conditions under which *ins-3* mutants showed increased resistance. Under these conditions, *hsf-1(sy441)* mutants did not exhibit increased sensitivity to oxidative stress and were even more resistant to heat stress compared to wild-type animals (S5 Fig). Our results are consistent with recent studies using similar heat stress conditions and developmental stages as those used in our study [47]. While we observed a reduction in oxidative stress resistance in *ins-3;hsf-1* double mutants compared to *ins-3* single mutants, this was not observed under heat stress conditions. In fact, the loss of *hsf-1* function did not affect the increased heat resistance of *ins-3* mutants (S5 Fig). These results suggest that DAF-16 activation appears to be the primary effector driving the enhanced stress resistance observed in these mutants. Our experiments, however, do not completely exclude a role for HSF-1 in modulating the responses to harsher stress conditions since *hsf-1* has been shown to play a critical role in stress resistance under prolonged stress conditions (e.g., thermotolerance after 9 h at 35 °C [48] or oxidative stress after 24 h of exposure to 200 mM paraquat [49]).

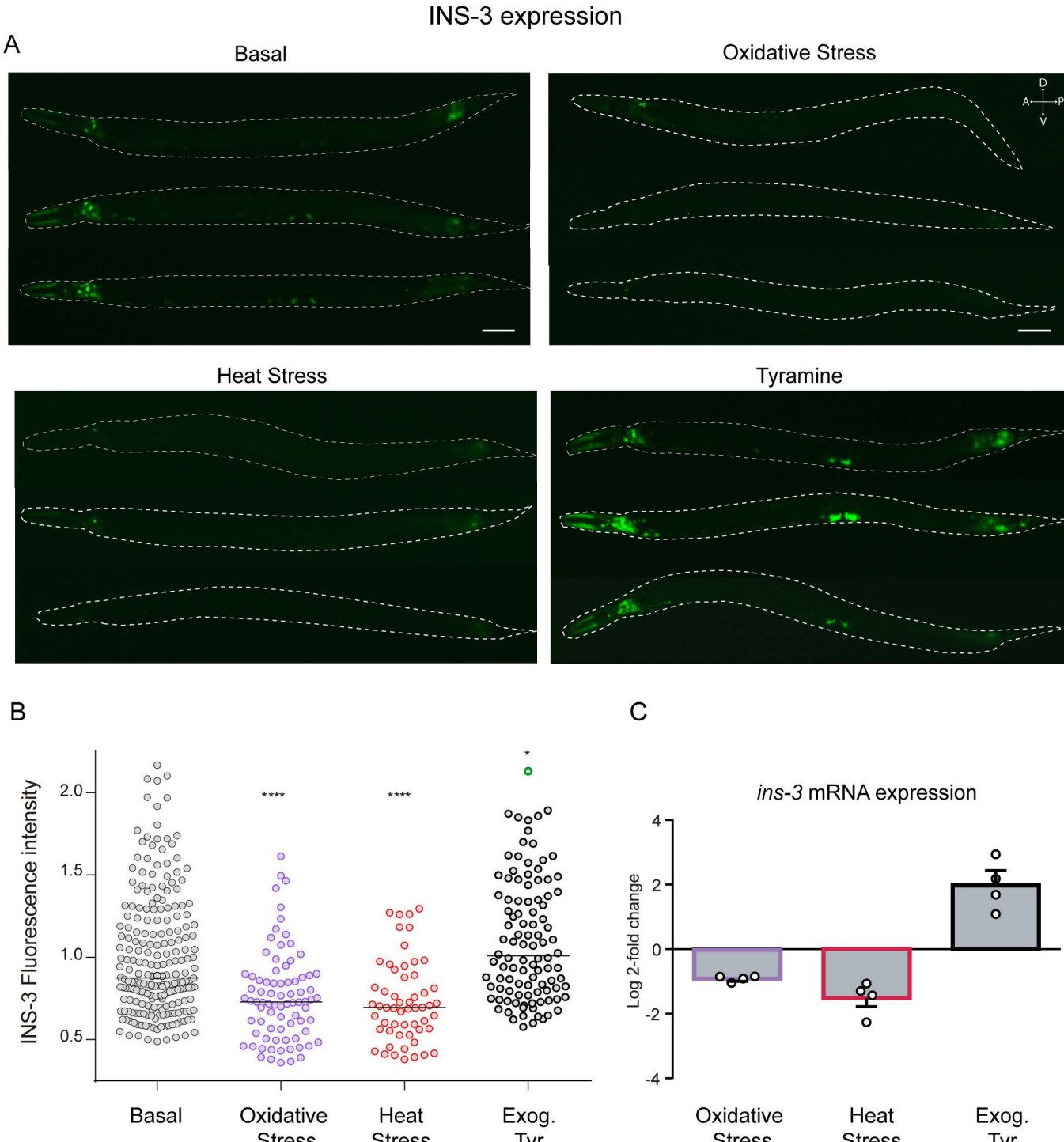

**Fig 4. Environmental stressors and tyramine have opposite effects on *ins-3* expression. (A)** Representative epifluorescence images (20×) of L4 animals expressing *Pins-3::GFP* under basal conditions (20 °C on Nematode Growth Media plates seeded with OP50), under oxidative stress (1 mM FeSO$_4$, 2 h) or heat stress (30 °C, 6 h) and in the presence of exogenous tyramine (15 mM). Scale bar 50 μm. **(B)** Corresponding quantification of fluorescence levels per worm. Scatter dot plot with relative expression of *Pins-3::GFP* normalized to basal conditions of each independent experiment. Line at the median. $n = 50$–200 animals per condition (distributed across 3–4 independent experiments). One-way ANOVA and Dunn's post hoc test vs. basal were used. $* p < 0.05$, $**** p < 0.0001$. **(C)** Log$_2$ fold-changes in *ins-3* transcript levels in animals exposed to oxidative stress (1 mM FeSO$_4$, 2 h) or heat stress (30 °C, 6 h) and in the presence of exogenous Tyramine (15 mM). Negative and positive values indicate down- and up-regulation of *ins-3*, respectively. Fold change was calculated as ΔCt basal conditions/ΔCt test conditions. Results are shown as mean ± s.e.m. $n = 4$ independent experiments. The data underlying this figure can be found at https://osf.io/wfgvs/.

## *ins-3* expression is differently modulated by distinct types of stressors

Exposure to different stressors can exert both positive and negative effects on the expression of ILP genes [24,50]. To investigate whether stressors affect *ins-3* expression, we exposed transgenic animals expressing a *Pins-3::GFP* reporter to oxidative and thermal stressors (Fig 4A, B). Exposure to either the oxidizing agent $FeSO_4$ (1 h) or heat (30 °C, 6 h), resulted in a decrease in green fluorescence protein (GFP) expression in both the intestine and neurons (Fig 4A, B). Consistent with these results, quantitative real-time PCR (qPCR) experiments also revealed lower *ins-3* mRNA levels in animals exposed to these stressors (Fig 4C). This decrease in expression upon exposure to these stressors does not appear to be a non-specific effect on protein expression and/or stability, as the expression of another ILP, *ins-4*, was unaffected in animals exposed to heat and even slightly increased upon oxidative stress (S6 Fig). In contrast to environmental stressors, exposure to exogenous tyramine significantly increases *ins-3* mRNA levels and enhances the expression of the transcriptional *Pins-3::GFP* reporter in the intestine (Fig 4). Since tyramine is released during the flight response [32,33], our data suggest that environmental stressors and the acute flight response have opposing effects on *ins-3* expression.

## Tyramine induces calcium transients in the intestine

The release of tyramine during the flight response stimulates the DAF-2/IIS signaling pathway by activating the $G_{\alpha q}$-coupled TYRA-3 receptor in the intestine [32]. We hypothesized that the tyraminergic activation of the $G_{\alpha q}$ signaling stimulates the release of intestinal ILPs. Activation of $G_\alpha q$ stimulates phospholipase C to generate diacylglycerol (DAG) and inositol trisphosphate (IP3), both of which have been shown to enhance insulin secretion in various organisms [51,52]. IP3 triggers $Ca^{2+}$ release from intracellular stores via the IP3 receptor intracellular storage [53,54]. Previous studies have demonstrated that the *C. elegans* IP3 receptor, inositol trisphosphate receptor 1 (ITR-1), regulates intestinal $Ca^{2+}$ oscillations that drive the defecation motor program (DMP) [55,56]. To analyze the molecular mechanisms downstream of TYRA-3 activation, we expressed the $Ca^{2+}$ indicator GCaMP6 in the intestine and measured $Ca^{2+}$ transients upon exposure to tyramine. On food, wild-type animals showed rhythmic $Ca^{2+}$ waves that propagated from the posterior to the anterior end of the intestine every 45–50 s coinciding with the DMP (Fig 5A, S1 Video). Tyramine deficient, *tdc-1* mutants and *tyra-3* mutants have a regular DMP and normal $Ca^{2+}$ waves similar to the wild-type (S7 Fig). In the absence of food, $Ca^{2+}$ waves and defecation events occur only very rarely in wild-type, *tdc-1*, and *tyra-3* animals consistent with previous observations [57]. Wild-type animals that were exposed to exogenous tyramine (30 mM) displayed a dramatic increase in $Ca^{2+}$ transients in the intestine, even in the absence of food (Fig 5B–D). Strikingly, tyramine also altered intestinal $Ca^{2+}$ dynamics, often producing high-frequency $Ca^{2+}$ waves that initiated from multiple locations along the length of the intestine and moved in both anterior and posterior directions (Fig 5B, S2 Video). While, in the absence of food, exogenous tyramine can induce rhythmic $Ca^{2+}$ waves and defecation behavior (S3 Video), most wild-type animals (53%) displayed high-frequency $Ca^{2+}$ waves (Fig 5E). Notably, in *tyra-3* null mutants, the tyramine-triggered intestinal $Ca^{2+}$ transients are significantly reduced (Fig 5B–E, S4 Video). Only 16% of *tyra-3* mutants displayed high-frequency waves in response to exogenous tyramine. Quadruple mutants of the tyramine receptors *lgc-55, ser-2, tyra-2*, and *tyra-3* were similar to *tyra-3* single mutants, with only 13% displaying high-frequency $Ca^{2+}$ waves in response to exogenous tyramine (S8 Fig). This suggests that TYRA-3 is the main tyramine receptor that stimulates $Ca^{2+}$ transients in the intestine. The tyramine-induced increase in intestinal $Ca^{2+}$ transients was largely abolished in *itr-1* mutants that lack the IP3 receptor ITR-1 (Fig 5B–E, S5 Video). Our data support the hypothesis that tyraminergic activation of the TYRA-3 receptor induces intestinal $Ca^{2+}$ transients through the activation of $G_{\alpha q}$-IP3 pathway.

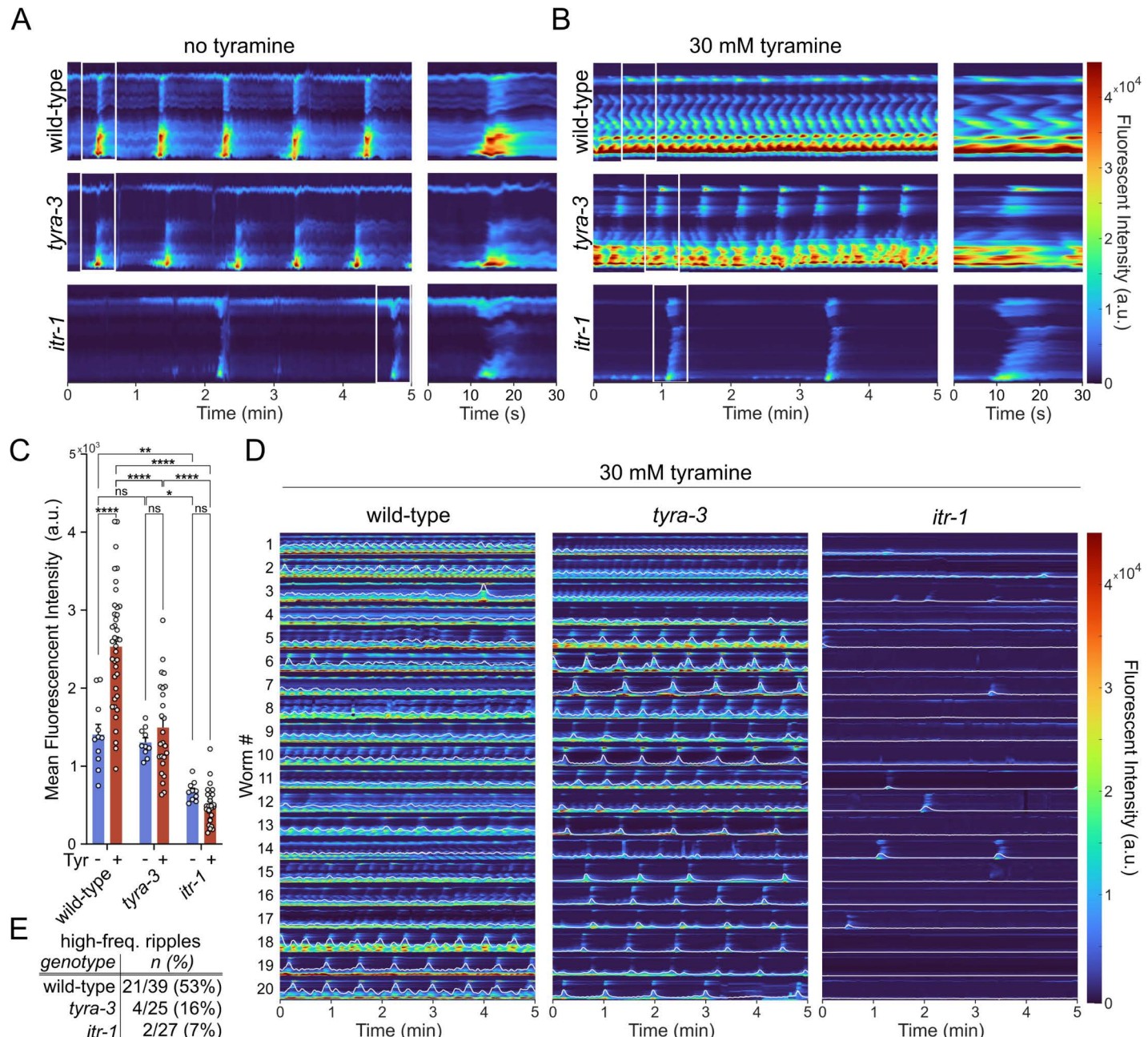

**Fig 5. Tyramine increases intestinal Ca²⁺ transients through the activation of TYRA-3. (A, B)** Representative kymographs of intestinal GCaMP fluorescence in freely behaving animals in the absence **(A)** or presence **(B)** of 30 mM tyramine. Kymographs are oriented with the anterior of the animal at the top. Insets at the right of each kymograph show an enlarged view of the Ca²⁺ waves indicated by the white rectangles. The color map values are at the right. a. u., arbitrary units. **(C)** Quantification of mean fluorescent intensity over 5-min recordings for wild-type, *tyra-3*, and *itr-1* mutants exposed to 0 mM (*n* = 10–11 animals) or 30 mM (*n* = 25–39 animals) tyramine. Results are mean ± s.e.m, Two-Way ANOVA with Tukey's multiple comparison test. ns, not significant. * $p < 0.05$, ** $p < 0.01$, *** $p < 0.001$, **** $p < 0.0001$. **(D)** Twenty stacked kymographs with an overlaid trace of mean fluorescent intensity (white lines) for wild-type, *tyra-3*, and *itr-1*, animals exposed to 30 mM tyramine, including the examples shown in panel B. The color map values for kymographs are shown at the right. a. u., arbitrary units. The *y*- axis range for each trace is −1,000 to 8,000 arbitrary units. **(E)** Table reporting the number of animals that exclusively displayed high-frequency Ca²⁺ waves without rhythmic Ca²⁺ waves. The data underlying this figure can be found at https://osf.io/wfgvs/.

## Tyramine triggers INS-3 release from the intestine

To determine whether tyramine stimulates the intestinal release of INS-3, we generated transgenic animals expressing the translational reporter *INS-3::VENUS* driven by the intestinal promoter *Pges-1*. Secreted VENUS-tagged peptides are taken up by the coelomocytes, which are scavenger cells that continually endocytose material secreted into the pseudocoelomic fluid. Fluorescence intensity in the coelomocytes can therefore be used as a proxy for peptide secretion [26,58,59]. Under basal conditions in transgenic *Pges-1::INS-3::VENUS* animals, we observed faint fluorescence in the intestine and the coelomocytes (Fig 6 and S9 Fig). This observation is consistent with a low tonic release of INS-3 from the intestine. Upon exposure to exogenous tyramine (15 mM), coelomocyte fluorescence intensity increased 2-fold (Fig 6). The increase in coelomocyte fluorescence is not due to a tyramine-induced increase in expression of *INS-3::VENUS* transgene driven by the *Pges-1* promoter, since qPCR showed no elevation in *Venus* mRNA levels (S10 Fig). Importantly, VENUS accumulation in coelomocytes following tyramine exposure was significantly reduced in *tyra-3* null mutants compared to wild-type animals (Fig 6). Therefore, our data indicate that tyraminergic activation of the TYRA-3 receptor stimulates INS-3 secretion from the intestine.

## Tyramine inhibits cytoprotective mechanisms through intestinal INS-3 secretion

Is INS-3 secretion required for the tyraminergic modulation of the stress response? To address this question, we examined the oxidative and thermal stress resistance of *ins-3* null mutants in the presence of exogenous tyramine. Unlike wild-type animals, exogenous tyramine did not impair the stress resistance of *ins-3* null mutants (Fig 7A). In addition, when we selectively rescued *ins-3* expression in the intestine, the negative impact of exogenous tyramine on stress resistance is restored to wild-type levels (Fig 7A).

To further investigate the crosstalk between tyramine and INS-3 in modulating the stress response, we examined the resistance of *ins-3;tdc-1* and *ins-3;tyra-3* double mutants. *These double mutants did not show any further enhancement in stress resistance compared to tdc-1, ins-3, or tyra-3 single mutants* (Fig 7B). This observation is consistent with the notion that

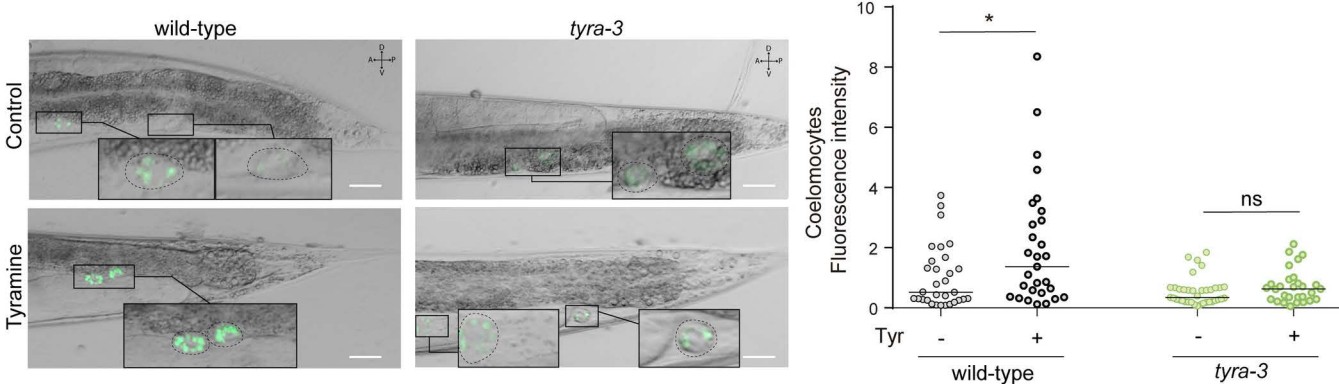

**Fig 6. Tyramine triggers INS-3 release from the intestine.** Representative epifluorescence images (40×) of young not gravid adults expressing *Pges-1::INS-3:: VENUS* in wild-type and *tyra-3* null mutant background, in the absence or presence of exogenous tyramine (15 mM). Scale bar 25 μm. Fluorescence is observed in the two posterior coelomocytes of worms (inset). Right. Corresponding quantification of fluorescence intensity on the coelomocytes (normalized to wild-type animals without tyramine exposure). Line at the median. $n = 35$–40 animals per condition distributed across three independent experiments. For conditions with tyramine, a two-tailed Student's *t* test vs. the same strain without tyramine was used. * $p < 0.05$. The data underlying this figure can be found at https://osf.io/wfgvs/.

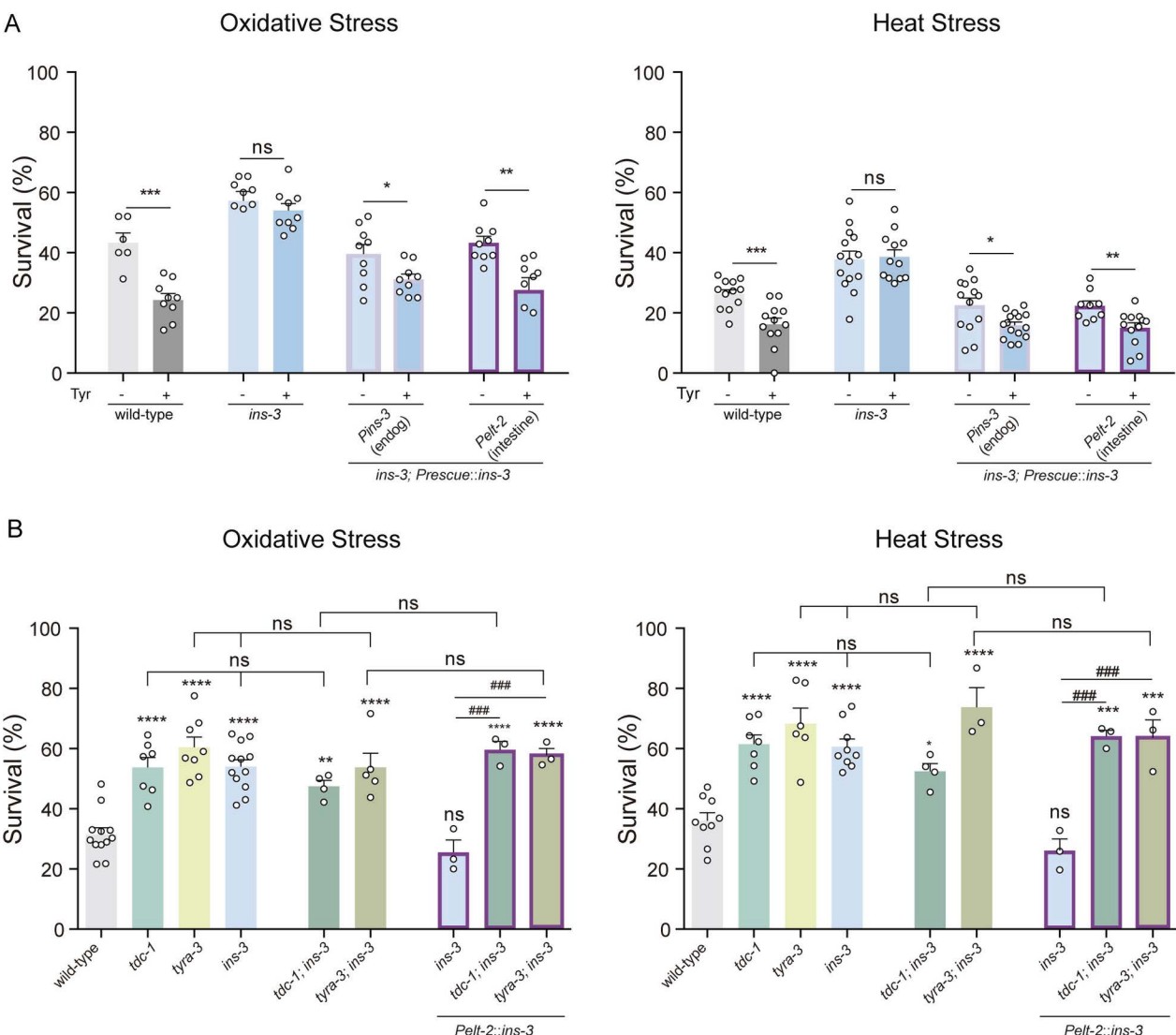

**Fig 7. Tyramine and INS-3 act in the same pathway to modulate stress resistance. (A)** Stress resistance of *ins-3* mutant animals expressing an *ins-3* cDNA driven by *Pins-3* (endogenous) or *Pelt-2* (intestinal) promoters upon exposure to oxidative stress (left) or heat stress (right) in the absence or presence of exogenous Tyramine (15 mM). The detrimental effect of exogenous tyramine on stress resistance is only abolished in *ins-3* mutant animals. Mean ± s.e.m. Six-fifteen independent experiments were performed (*n* = 6–15). Each experiment included 25–30 worms per condition. For conditions with tyramine, two-tailed Student's *t* test vs. the same strain without tyramine was used. ns, not significant. * *p* < 0,05 ** *p* < 0.01, *** *p* < 0.001. **(B)** Survival percentages of animals exposed to oxidative (left) and heat stress (right). Results are shown as mean ± s.e.m. Three to twelve independent experiments were performed (*n* = 3–12). Each experiment included 40–50 animals per condition. The stress resistance is not further enhanced in double mutants *tdc-1;ins-3* and *tyra-3;ins-3* compared to the single mutants. Additionally, the intestinal expression of *ins-3* does not restore resistance to wild-type levels in tyramine-deficient backgrounds. One-way ANOVA, Holm–Sidak's post hoc test vs. wild-type were used (**p* < 0.05, ***p* < 0.01, ****p* < 0.001, *****p* < 0.0001). One-way ANOVA, Dunnett's post hoc test vs. intestinal rescue of *ins-3* were used ( ###*p* < 0.001). One-way ANOVA, Dunnett's post hoc test were used for mutants deficient in both tyramine and *ins-3* (*ins-3*;*tdc-1* and *ins-3*;*tyra-3*) compared to the corresponding single mutants. ns, not significant. The data underlying this figure can be found at https://osf.io/wfgvs/.

*tdc-1*, *tyra-3*, and *ins-3* genes act in the same pathway to regulate the environmental stress response. Unlike our observations in *ins-3* null mutant backgrounds, the intestinal rescue of *ins-3* in *ins-3;tdc-1 and ins-3;tyra-3* backgrounds did not impact the stress resistance (Fig 7B). Furthermore, exogenous tyramine failed to decrease stress resistance in the intestinal *ins-3*

rescue strain in *tyra-3* null mutant background (S11 Fig). These findings strongly support the hypothesis that INS-3 acts downstream of tyramine to modulate the stress response.

Since nuclear translocation of DAF-16 is enhanced in *ins-3* null mutant animals under mild stress (Fig 3A), we analyzed the effects of exogenous tyramine on DAF-16 localization in these mutants. After strong heat (35 °C, 30 min), DAF-16/FOXO predominantly localized to the nucleus in both wild-type and *ins-3* mutants [32] (Fig 8A). Exogenous tyramine reduced DAF-16/FOXO nuclear localization in wild-type animals, consistent with previous reports [32]. The addition of exogenous tyramine; however, failed to inhibit DAF-16 nuclear translocation in *ins-3* mutant animals (Fig 8A). Intestinal expression of *ins-3* restores the inhibitory effect of tyramine on the nuclear translocation of DAF-16/FOXO. This indicates that the negative modulation of DAF-16 by tyramine relies on the release of INS-3 from the intestine (Fig 8A). Taken together, our findings strongly support a model in which the flight neurohormone tyramine activates the TYRA-3 receptor in the intestine, stimulating the release of INS-3 from the intestine; INS-3 in turn, activates the DAF-2 pathway in various tissues, thus inhibiting cytoprotective mechanisms (Fig 8B).

## Discussion

The insulin/insulin-like growth factor signaling (IIS) pathway plays a crucial role in the stress response throughout the animal kingdom [7,60]. Previous studies have shown that downregulation of insulin signaling in *C. elegans* increases life span, and improves resistance to different environmental stressors [15,18,61–63]. While key components of the IIS pathway, such as the DAF-2 receptor and downstream molecules are well characterized, the specific roles of individual ILPs in the stress response remain largely unclear. The *C. elegans* genome contains 40 putative ILPs [64], with dynamic temporal and spatial expression patterns with agonist or antagonist impact on the DAF-2 receptor. This complexity indicates the importance of the DAF-2/IIS pathway regulation in the animals' adaptation to environmental changes. Interestingly, most of the ILPs in the worm are expressed in the intestine [24]. Like other animals, the *C. elegans* intestine not only serves essential functions in digestion but also plays a pivotal role in the stress response [65]. Previous studies have shown that intestinal DAF-2/IIS is crucial for regulating resistance to oxidative stress, as intestinal-specific deletion of DAF-2 leads to increased oxidative stress resistance in a DAF-16-dependent manner [66]. In addition, the intestine also plays a non-cell autonomous role in the DAF-2-dependent control of life span [67–69].

We find that animals lacking the ILP encoding gene, *ins-3*, have an increased resistance to oxidative stress, heat, and starvation. Our data show that INS-3 activates the DAF-2/IIS pathway, and thus inhibits the environmental stress response. While *ins-3* is expressed in the nervous system and the intestine of *C. elegans*, intestinal *ins-3* expression is sufficient to restore oxidative and heat stress sensitivity to wild-type levels. Our findings indicate that INS-3 release from the intestine plays a key role in modulating systemic environmental stress response, contributing to the broader network of stress regulation in *C. elegans*. Importantly, the production of ILPs in the intestine may not be unique to *C. elegans*, as the ability to express and secrete insulin by epithelial cells of the intestine has been reported in rats [70,71]. Furthermore, similar to our observations, this expression appears to be modulated by stressful conditions [72].

While there is increasing recognition of the importance of brain–gut communication in animal physiology and behavior, the molecular mechanisms underlying this interaction remain poorly understood. In previous work, we identified a novel brain–gut communication pathway that sheds light on how the repeated activation of the flight response can impair

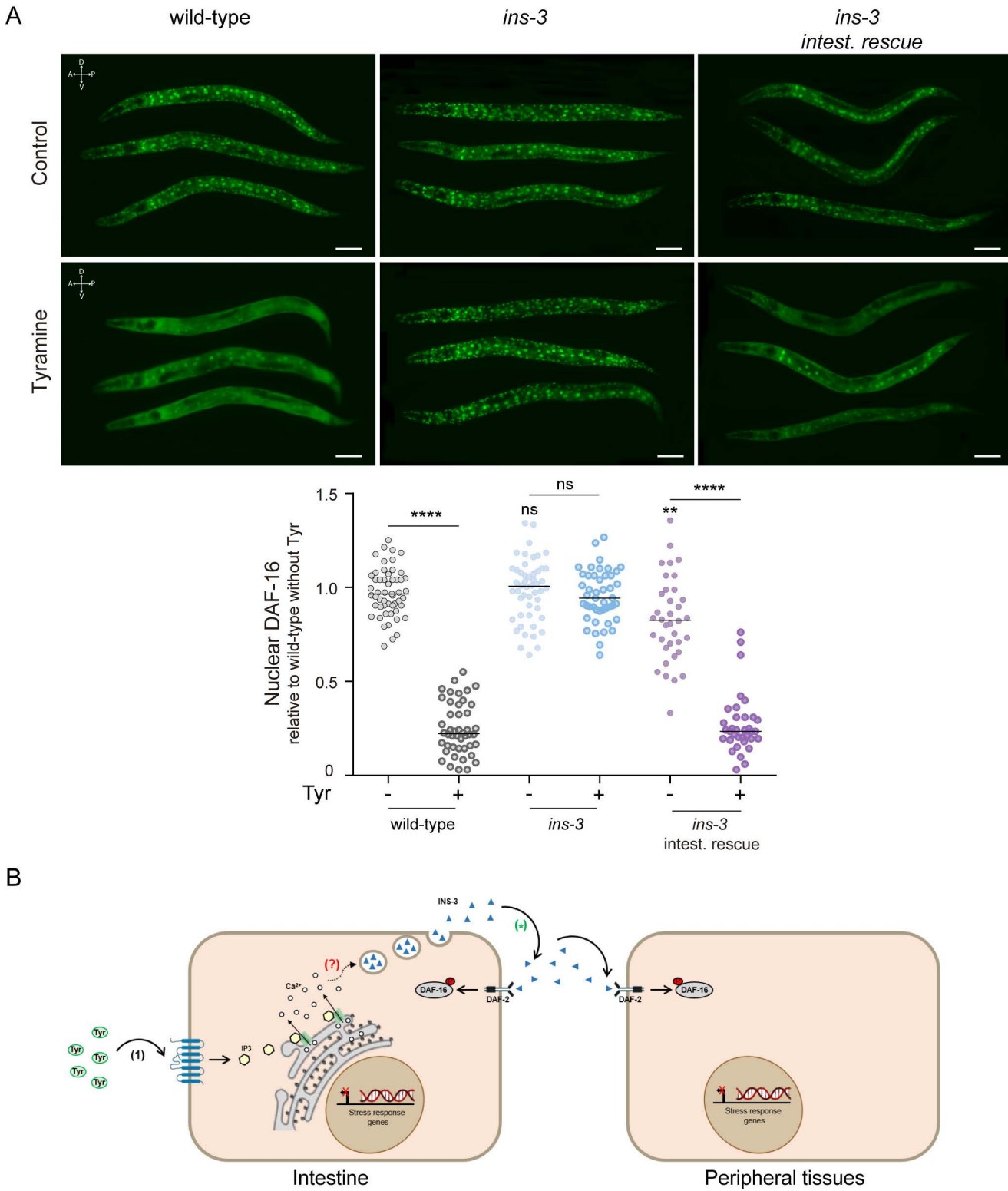

**Fig 8. Tyramine inhibits cytoprotective mechanisms through intestinal INS-3 secretion. (A)** Top: DAF-16a/b::GFP localization upon strong heat (35 °C) for 30 min, in the absence or presence of exogenous tyramine (15 mM). Scale bar, 50 μm. Tyramine can inhibit DAF-16 nuclearization upon expression of *ins-3* in the intestine. Bottom: Corresponding scatter dot plots with the number of cells with nuclear DAF-16 per animal (normalized to wild-type worms without tyramine; line shows median). *n* = 35−50 animals per condition distributed across three independent experiments. For conditions without tyramine, One-way ANOVA with Dunn's multiple comparisons post hoc test vs. the wild type was used. ns, not significant, **$p < 0.01$. Two-tailed

Student's *t* test (no tyramine vs. tyramine) was used. ns, not significant. ****$p < 0.0001$. **(B)** Model: The escape neurohormone tyramine activates TYRA-3 in the intestine, elevating calcium levels, likely through IP3-dependent mechanisms involving ITR-1. Activation of TYRA-3 triggers the intestinal release of INS-3, which systemically activates DAF-2. DAF-2 activation results in the phosphorylation of DAF-16, preventing its nuclear translocation and inhibiting the transcription of cytoprotective genes. References: (1) signaling pathway previously found in [32], (?) unknown regions of the pathway; (*) signaling pathway explained in this article. The data underlying this figure can be found at https://osf.io/wfgvs/.

the reaction to other environmental challenges [32]. Tyramine, considered the counterpart of adrenaline in *C. elegans*, is synthesized through the decarboxylation of the amino acid tyrosine by the enzyme tyrosine decarboxylase-1 (TDC-1). This enzyme is expressed in the tyraminergic RIM neurons, octopaminergic ring interneuron CUV1 (RIC) neurons, and neuroendocrine uterine-vulval cell 1 (UV1) cells [33]. Despite its expression in these various cell types, we previously demonstrated that tyramine release from RIM neurons during the flight response activates the TYRA-3 receptor in the intestine [32]. TYRA-3 activation, in turn, leads to the systemic stimulation of the DAF-2/IIS pathway and the inhibition of cytoprotective transcription factors, such as DAF-16/FOXO [32]. In this study, we show that the release of tyramine activates TYRA-3, stimulates the secretion of INS-3 from the intestine, and inhibits DAF-2-dependent cytoprotective mechanisms. Phylogenetic analyses indicate that TYRA-3 clusters with other G-protein-coupled receptors that are coupled to $G_q$ signaling pathways [73]. $G_q$ signaling stimulates DAG and IP3 production. We show that tyramine can dramatically increase $Ca^{2+}$ transients in the intestine of *C. elegans*, a process modulated by the TYRA-3 receptor and dependent on the IP3 receptor ITR-1. Interestingly, studies in mammals have shown that the activation of $G_q$ receptors and the consequent triggering of DAG- and IP3-dependent mechanisms are essential for insulin secretion [74,75]. DAG activates protein kinase C and opens T-type $Ca^{2+}$ channels [76]. IP3 stimulates the IP3 receptor, increasing $Ca^{2+}$ levels from intracellular stores [53,54]. These findings suggest a potential parallelism between the $G_q$-mediated pathways involved in insulin secretion in mammals and the TYRA-3 signaling pathway in *C. elegans*. Additional experiments are required to unravel the molecular mechanisms by which activation of a $G_q$-coupled receptor leads to the release of dense core vesicles containing ILPs. The tyramine-mediated secretion of INS-3, provides a novel molecular brain–gut communication pathway that links the flight response with the suppression of cytoprotective mechanisms.

Although the activation of cytoprotective pathways is crucial to protect against the potential costs of cellular damage, repair, and death, it comes at the expense of resources that could otherwise be allocated to animal development and reproduction [77,78]. This is consistent with our observations that stress-resistant *ins-3* mutants, exhibit a slower developmental rate and a reduced brood size. It is likely advantageous for an animal to limit the activation of cytoprotective pathways under favorable conditions, while still retaining the capacity to induce these pathways in emergencies. This delicate balance between maintaining normal reproductive development while preserving inducible cytoprotective capacity is crucial for the survival and overall fitness of an organism. The neural perception of stressors and subsequent regulation of cytoprotective responses play a pivotal role in this plasticity, ensuring that resources are allocated optimally to support both growth and defense mechanisms.

In vertebrates, the release of adrenaline (the vertebrate counterpart of tyramine) in response to acute stressors is a fundamental component of the flight response [79,80]. Frequent exposure to acute stressors can lead to chronic activation of the stress hormone system [81,82]. Similar to our findings in *C. elegans*, this chronic activation weakens the animal's defense systems against other stressors, compromising its ability to cope with subsequent challenges [83,84]. In humans, persistent exposure to real or perceived danger, as seen in conditions like stress disorders, has been associated with adverse health outcomes such as

type 2 diabetes, increased oxidative stress, neurodegenerative disorders, and premature death [85–90]. Individuals with stress disorders often exhibit hyperinsulinemia [91–94] emphasizing the potential role of insulin dysregulation in stress-related health impairments. The precise mechanisms by which increased levels of stress neurohormones impair general health and defense mechanisms in humans are still not fully understood. However, considering the remarkable conservation of neural control over stress responses across species, it would be interesting to investigate whether chronic release of adrenaline negatively impact health by stimulating the secretion of ILPs.

## Materials and methods

### Strains and maintenance of *C. elegans*

Standard *C. elegans* culture and molecular biology methods were used [95,96]. All *C. elegans* strains were grown at 20 °C on Nematode Growth Medium (NGM) agar plates with OP50 *Escherichia coli* as a food source. Low population density was maintained throughout the development and during the assays. The wild-type reference strain used in this study is N2 Bristol. Some of the strains were obtained through the *Caenorhabditis* Genetics Center (CGC, University of Minnesota), which is funded by the NIH Office of Research Infrastructure Programs (P40 OD010440). All strains used in this study were backcrossed four times with N2 to eliminate possible background mutations.

The strains used were:

- N2 (wild type) [95]

- OAR186 RB1915 *ins-3 (ok2488)* II 4xBC

- MT10661 *tdc-1(n3420)* II [33]

- CX11839 *tyra-3(ok325)* X [97]

- HT1690 *unc-119(ed3)* III*; wwIs26 [Pins-3::GFP + unc-119(+)]* [24]

- HT1693 *unc-119(ed3)* III*; wwEx63 [Pins-4::GFP + unc-119(+)]* [24]

- OAR45 *nbaEx8 [Pins-3::INS-3 (20)+lin-15 (80)];ins-3(ok2488), lin-15 (n765ts)]*

- OAR43 *nbaEx6 [Pelt-2::INS-3 (20)+lin-15 (80)];ins-3(ok2488), lin-15 (n765ts)]*

- OAR50 *nbaEx13 [Prgef-1::INS-3 (20)+lin-15 (80)];ins-3(ok2488), lin-15 (n765ts)]*

- OAR11 *tdc-1 (n3420)* II*; ins-3 (ok2488)* II

- OAR169 *tyra-3 (n3420)* X*; ins-3 (ok2488)* II

- OAR68 *nbaEx6 [Pelt-2::INS-3 (20)+lin-15 (80)]; tyra-3(ok325)* X*; ins-3 (ok2488)* II*; lin-15 (n765ts)]*

- OAR69 *nbaEx6 [Pelt-2::INS-3 (20)+lin-15 (80)]; tdc-1(n3420)* II*; ins-3 (ok2488)* II*; lin-15 (n765ts)]*

- QW2436 *Pges-1::INS-3::VENUS*

- TJ356 *zIs356[Pdaf-16::DAF-16a/b::GFP + pRF4]* [98]

- OAR58 *zIs356[Pdaf-16::DAF-16a/b::GFP + pRF4]; ins-3(ok2488)* II

- OAR73 *zIs356[Pdaf-16::DAF-16a/b::GFP + pRF4]; nbaEx6[Pelt-2::INS-3 (20)+lin-15]; ins-3(ok2488)* II*; lin-15(n765ts)]*

- CL2070 *dvIs70[Phsp-16.2::GFP + pRF4]* [46]

- OAR100 *dvIs70[Phsp-16.2::GFP + pRF4]; ins-3(ok2488)* II
- OAR158 *dvIs70[Phsp-16.2::GFP + pRF4]; ins-3(ok2488)* II; *nbaEx6[Pelt-2::INS-3 (20)+lin-15];ins-3(ok2488)*II, *lin-15(n765ts)* X
- CB1370 *daf-2(e1370)* III [99]
- GR1307 *daf-16(mgDf50)* I [100]
- *PS3551 hsf-1(sy441)* I
- *OAR56 ins-3 (ok2488)* II; *daf-2(e1370)* III
- *OAR99 ins-3 (ok2488)* II; *daf-16(mgDf50)* I
- *OAR194 ins-3 (ok2488)* II; *hsf-1(sy441)* I
- *VP303* rde-1(ne219) *V*; *kbIs7[nhx-2p::RDE-1 + rol-6(su1006)]* [55]
- *OAR195 ins-3(ok2488)* II; rde-1(ne219) *V*; kbIs7*[nhx-2p::RDE-1 + rol-6(su1006)]*
- *OAR196 tyra-3(ok325)* X*; Pges-1::ins-3::venus*
- QW2323 *zfIs178[Pges-1::NLSwCherry::SL2::GCaMP6s::unc-54 UTR (80ng/ul); lin-15(+) (80ng/ul)]* I 4x o.c.
- QW2331 *zfIs178[Pges-1::NLSwCherry::SL2::GCaMP6s::unc-54 UTR (80ng/ul); lin-15(+) (80ng/ul)]* I; *tyra-3(ok325)* X
- QW2335 *zfIs178[Pges-1::NLSwCherry::SL2::GCaMP6s::unc-54 UTR (80ng/ul); lin-15(+) (80ng/ul)]* I; *itr-1(sa73)* IV

## RNAi silencing

RNAi was carried out by feeding nematodes with dsRNA-producing bacteria as described previously [101]. RNAi clones that target *C. elegans* insulin genes were obtained from Ahringer's feeding RNAi library [102]. RNAi plates were prepared with standard NGM agar supplemented with 25 mg/ml ampicillin and 1 mM isopropyl b-D-1-thiogalactopyranoside, poured into 35 mm plates and allowed to dry for 7 days at 4 °C. Fresh HT115 *E. coli* bacteria transformed with the L440 empty vector or carrying the appropriate RNAi clone were grown in 2 mL of LB containing 100 mg/ml ampicillin at 37 °C overnight. Overnight bacterial culture was centrifuged for 2 min at 3,500 rpm and resuspended into 500 µL of LB. Seventy-five microliters of this resuspension were seeded on RNAi plates, and plates were kept overnight at 20 °C for the induction of RNAi.

Animals in the fourth larval stage (L4) were transferred to each plate and maintained at 20 °C. Once they laid eggs, these animals were removed. Therefore, oxidative stress assays were performed in animals (L4 stage) hatched and grown in RNAi plates.

## Stress resistance assays

**Oxidative stress.** Iron sulfate (FeSO$_4$) was used as an oxidative stressor as previously described [32,103]. Twenty L4 animals were transferred to 35 mm agar plates (4–5 plates) containing FeSO$_4$ at the indicated concentration and time (1 mM, 2 h for microscopy and 15 mM, 1 h for survival assays). To analyze the effect of tyramine on stress resistance, L4 animals were transferred to four 35 mm NGM agar plates (containing 30–40 animals each) seeded with OP50 bacteria with or without 15 mM exogenous tyramine. Stress assays were performed after 14 h.

**Heat stress.** Thermotolerance assays were performed as described [32]. For each strain, four 35 mm NGM agar plates containing 20 animals (with or without 15 mM exogenous tyramine) were incubated at 35 °C for 4 h. To ensure proper heat transfer, 6-mm-thick NGM agar plates were used. Animals were synchronized as L4s and used 14 h later. Plates were sealed with Parafilm in zip-lock bags and immersed in a 35 °C water bath for 4 h. After that period, the plates were taken out from the water bath and the zip-lock bags and parafilm were removed. Surviving animals were counted after a 20 h recovery period at 20 °C. For all assays, animals were scored as dead if they failed to respond to prodding with a platinum-wire pick to the nose of the animal.

**Starvation resistance.** Food deprivation resistance assays were performed as described [32]. L4 animals were rinsed off the plate, washed with M9 buffer, and then seeded in a 96-well multiwell plate (one worm per well) containing M9. Survival was monitored on days 4, 6, and 8 from the day food was removed. Animals with larvae in their uterus ("bag of worms") were occasionally observed and were excluded from the quantification.

## Microscopy and image analysis

For microscopy, age-synchronized animals were mounted in M9 with levamisole (10 mM) onto slides with 2% agarose pads. *ins-3* expression pattern was analyzed by confocal microscopy as previously described [32]. Images were acquired on confocal microscopy (LSCM; Leica DMIRE2) with 20× and 63× objectives. For *ins-3 and ins-4* expression levels analysis, animals containing the corresponding transcriptional GFP reporter (*Pins-3::GFP* and *Pins-4:GFP*, respectively) were imaged using an epifluorescence microscope (Nikon Eclipse TE2000-5) coupled to a CCD camera (Nikon DS-Qi2) with 20× objective. Fluorescence intensity was quantified using a very simple Macro that runs on Image J FIJI software. With this Macro, the background was initially subtracted, and then a region of interest (ROI) was selected (i.e., in the posterior region of the intestine). The mean fluorescence intensity of this selected region was measured. The instructions for creating this macro are available at OSF (https://osf.io/wfgvs/).

## Tyramine supplementation

Tyramine hydrochloride (Alfa Aesar) stocks were made with MilliQ sterile water and diluted to 15 mM into NGM agar before pouring. For calcium imaging experiments, tyramine was diluted to 30 mM in agar. Plates were stored at 4 °C and used within 1 week after pouring.

## Reverse transcription-quantitative polymerase chain reaction (RT-qPCR)

Total RNA was extracted from approximately 100 adult worms using Bio-Zol RNA isolation reagent (PBL-Productos Bio-Logicos). To lyse the worms, five cycles of cold–warm shock were applied, alternating between liquid nitrogen and a 35 °C water bath. Following RNA extraction, the RNA was reverse-transcribed into complementary DNA (cDNA) using Easy-Script Reverse Transcriptase (TransGen Biotech).

Gene expression analysis was conducted using qPCR on a Rotor-Gene 6000 (Corbett Research) with an SYBR Green Master Mix. The cycling conditions for qPCR were set as follows: an initial incubation at 50 °C for 2 min, followed by denaturation at 95 °C for 2 min, and then 45 cycles of 15 s at 95 °C, 30 s at 60 °C, and 30 s at 72 °C. A melting curve analysis was performed after the final cycle to assess the specificity of the amplified products in each reaction.

The expression level of each gene was quantified relative to that of actin, used as the housekeeping gene control, with $\Delta CT$ calculated as $\Delta CT = CT_{\text{Gene of interest}} - CT_{\text{actin}}$. The transcript

levels of the gene of interest for each condition were assessed relative to the control condition using the formula ($\log_2 = (\Delta CT_{control} / \Delta CT_{Test\ condition})$). Positive values indicate an increase in expression levels, while negative values indicate a decrease compared to the control condition. Values of Ct above 35 are below the detection limit and considered background noise. All experiments were performed at least as three independent assays. For each assay, we run three technical replicates of each PCR reaction. Primers were designed using Integrated DNA Technologies online design tools.

Primer sequences for RT-qPCR

- *ins-3* **(Forward):CCTGATGGCCAGATCAAGAA**

- *ins-3* **(Reverse):ACATTCTCCTCCACACATTACC**

- *venus* **(Forward):ACAGCCACAACGTCTATATCAC**

- *venus* **(Reverse):TTGTACAGCTCGTCCATGC**

- *hsp-16.2* **(Forward):ACCACTATTTCCGTCCAGCT**

- *hsp-16.2* **(Reverse):GGCGTTCCATCAGAGCCAT**

## Calcium imaging

Young adult animals expressing the integrated transgene *zfIs178[Pges-1::NLSwCherry::SL2:: GCaMP6s]* were transferred to agar plates containing 0 or 30 mM tyramine and allowed to acclimate for 5 min prior to imaging with a 5× objective (Zeiss, Fluar 5×/0.25) on an inverted fluorescent microscope (Zeiss, Axio observer A1) equipped with an EGFP/mCherry optical splitter (Cairn, OptoSplit II). The Chroma 59022 ET filter set was used in the microscope and Chroma filters ET525/50m, ET632/60m, and dichroic T560lpxr were used in the optical splitter. Transmitted light provided a brightfield image in the mCherry channel that was used for analysis. Recordings were acquired at 15 frames per second with a 66-ms exposure time on an ORCA-Flash4.0 Digital CMOS camera (Hamamatsu, C13440-20CU) using μManager software [104] with the Split View plugin to generate an image stack with interleaved brightfield and GFP channels.

## Intestinal calcium measurement

Custom MATLAB (The MathWorks) scripts were used to measure intestinal GCaMP fluorescence. Brightfield images were thresholded using Otsu's method to generate a binary mask which was refined through morphological erosion and dilation and size filtering. A ROI was generated from the processed binary mask which was used to measure pixel values in the GCaMP channel. A background subtracted mean fluorescent signal was calculated for each frame by subtracting the mean gray value of pixels outside of the ROI from the mean gray value of pixels inside the ROI. To generate kymographs of GCaMP signal, a midline was created via skeletonization of the ROI. At each pixel of the midline, the neighboring pixels were used to create a line segment. The maximum gray value was measured along a line perpendicular to each line segment and the mean background signal was subtracted to yield GCaMP fluorescence along the length of the worm. Linear interpolation was used to ensure a consistent number of midline samples to plot kymographs. Quantification of rhythmic $Ca^{2+}$ waves and high-frequency ripples was done by visual inspection of the kymographs and mean fluorescent signal traces. Rhythmic $Ca^{2+}$ waves were identified by two or more peaks that rose above baseline fluctuation in the mean fluorescent signal. High-frequency waves were identified by local oscillations in the kymograph that have a period of less than 30 s. Animals were scored as

displaying high-frequency Ca$^{2+}$ waves if local oscillations were present and rhythmic propagating calcium waves were not detectable.

## INS-3 release quantification

A population of animals expressing the *Pges-1::INS-3::VENUS* transgene was exposed to tyramine (15 mM, 14 h). Venus is resistant to quenching in low pH environments and has been extensively used to monitor peptide secretion from dense core vesicles in *C. elegans* [105–107]. Individuals were immediately mounted and anesthetized as described above. Images were acquired on confocal microscopy (LSCM; Leica DMIRE2). Images were loaded into ImageJ and the soma of the coelomocytes were identified from their anatomical location. A circular ROI was placed around the coelomocyte to measure the mean fluorescence intensity value of each cell. The values obtained for each condition were normalized to the average intensity of the coelomocytes in the control condition.

## Subcellular DAF-16 localization

DAF-16 translocation was analyzed using strains containing the translational P*daf-16::DAF-16A/B::GFP* reporter in a wild type, or insulin mutant background (P*daf-16::DAF-16/B::GFP;ins-3*; *Pdaf-16::DAF-16A/B::GFP;ins-3; P*elt-2::INS-3*). Young adult, not gravid animals selected after the last molt (at least 25 animals per experiment, repeated 3–4 times) in basal conditions or exposed to either mild- (35 °C, 10 m) or strong-heat stress (35 °C, 30 m), with or without exogenous tyramine, were mounted to analyze DAF-16 cellular distribution under a fluorescence microscope. The number of GFP-labeled nuclei per animal was quantified using Image J FIJI software and normalized to the wild-type strain without tyramine condition within the same day.

## Expression analysis of DAF-2–IIS target genes

*hsp-16.2* expression levels were analyzed in transgenic strains containing transcriptional reporters in the wild-type and *ins-3* mutant backgrounds. Animals were imaged using an epifluorescence microscope (Nikon Eclipse E-600) coupled to a CCD camera (Nikon K2E Apogee). Fluorescence intensity was quantified using Image J FIJI software.

## Molecular biology and cloning

Standard molecular biology techniques were employed for the endogenous rescue of *ins-3*. The entire *ins-3* gene, along with a 2.1 kb upstream region from the start codon, was amplified from genomic DNA. Subsequently, this fragment was cloned into a vector backbone derived from the plasmid Ppd95.75, utilizing the *unc-54* 3'UTR. For the construction of intestinal and neuronal rescue constructs, the *ins-3* genomic DNA (including intronic sequences) from this construct was subcloned behind the intestinal reporters *elt-2* and the pan-neuronal *rgef-1* promoter, respectively. Since we did not observe changes in stress resistance in *Prgef-1::INS-3* rescued animals compared to the *ins-3* null mutant, we performed qPCR to ensure that *ins-3* is effectively expressed (S1 Table). Additionally, the *ins-3* gene was subcloned into a plasmid containing the intestinal promoter *ges-1* and the YFP variant Venus [58] to generate the reporter *Pges-1::INS-3::VENUS*. The sequences of the constructed plasmids are available on the Open Science Framework platform (https://osf.io/wfgvs/). Transgenic strains were generated through the microinjection of plasmid DNA at 20 ng/µl into the germline, along with the co-injection marker *lin-15* rescuing plasmid pL15EK (80 ng/µl), into *lin-15(n765ts)* mutant animals. For the intestinal expression of *INS-3* and *INS-3::VENUS*, the plasmid DNA concentration was 3 ng/µl, since higher concentrations did not yield transgenic animals (possibly

due to toxic effects of high INS-3 levels in the intestine). At least three independent transgenic lines were established, and the data presented are from a single representative line.

Double mutants were obtained using standard techniques [108]. In brief, 10 males from one of the mutant strains of interest (*tdc-1* or *tyra-3*) were mated with 2 *ins-3* null mutant hermaphrodites. After 72 h, F1 worms (one per plate) were isolated to allow for the production of offspring. The resulting F2 generation was isolated again. The double mutants were identified by PCR analysis of genomic DNA.

### Data collection and statistics

All data are represented in a format that shows the distribution (dot plots) and all the graph elements (median, error bars, etc.) are defined in each figure legend. For most of our experiments, as is usual in *C. elegans* research, we used a large number of animals per condition in each assay (typically more than 40–50 animals). This number of animals is large enough to ensure appropriate statistical power in the tests used. All the statistical tests were performed after checking normality. Grubbs' test was used for outliers analysis ($p < 0.05$). We performed the experiments at least 3–4 times to ensure reproducibility. All the animals were grown in similar conditions and the experiments were performed on different days, with different animal batches. In general, the experimenter was blind to the conditions/strains tested. Drugs were previously controlled by analyzing a known phenotype (e.g., worm paralysis and head relaxation on 30 mM exogenous tyramine). Animals used were age-synchronized (L4 or 14 h past L4).

### Supporting information

**S1 Fig. *ins-3* null mutants are more resistant to starvation than wild-type animals.** Survival percentages of wild-type and *ins-3* null mutant worms upon 4, 6, and 8 days of starvation. Three independent experiments were performed $n = 3$. Each experiment included 45–50 animals per condition. Two-tailed Student's *t* test was used. * $p < 0.05$. The data underlying this figure can be found at https://osf.io/wfgvs/.
(JPG)

**S2 Fig. DAF-16 primarily localizes to the cytoplasm in non-stressed *ins-3* null mutants.** Representative fluorescence images (20×) depicting the localization of DAF-16a/b::GFP under basal conditions in wild-type, *ins-3* null mutant, and intestinal rescue of *ins-3* backgrounds. No nuclear localization of DAF-16 was observed. $n = 20$–30 animals per condition. Scale bar, 50 µm.
(JPG)

**S3 Fig. *ins-3* mutants have a reduced brood size and develop more slowly than wild-type animals.** Left. Total number of progeny per worm in wild-type and *ins-3* null mutants. $n = 25$–30 animals per condition distributed across (three) independent experiments. Two-tailed Student's *t* test was used. ** $p < 0.01$. Right. Developmental rate of wild-type and *ins-3* null mutant worms. A color code was used to represent each of the following animal stages: L1–L3: early larval stages, L4: last larval stage, > L4: adult stage. The animal classification was evaluated at the indicated time points (24, 36, 48, 60, and 72 h). Five to six independent experiments were performed ($n = 5$–6). Each experiment included 100–200 worms per condition Data are represented in a stacked bar chart as mean. A two-tailed Student's *t* test was used. * $p < 0.05$. The data underlying this figure can be found at https://osf.io/wfgvs/.
(JPG)

**S4 Fig. *hsp-16.2* expression is enhanced in *ins-3* mutants. (A)** Representative fluorescence images (20×) of animals expressing *Phsp16.2::GFP* in wild-type, *ins-3* null mutants, and

intestinal rescue of *ins-3* backgrounds under basal conditions and after 15 min of heat exposure (35 °C) followed by 70 min recovery at 20 °C. Scale bar, 50 μm. Right. Corresponding quantification of the fluorescence level per animal. Scatter dot plot with relative expression of *Phsp16.2::GFP* (normalized to wild type of each independent experiment) (line shows median). $n = 10$–20 animals per condition distributed across three to four independent experiments. One-way ANOVA and Dunn's post hoc test for multiple comparisons among groups in basal conditions were used. ns, not significant. One-way ANOVA and Dunn's post hoc test for multiple comparisons among groups after heat stress were used **$p < 0.01$, ****$p < 0.0001$. Two-tailed Student's *t* test (basal vs. heat stress) was used. ****$p < 0.0001$. **(B)** Log$_2$ fold-changes in *hsp-16.2* transcript levels in animals exposed 15 min of heat exposure (35 °C). Negative and positive values indicate down- and up-regulation of this gene compared to wild-type animals, respectively. Fold change was calculated as ΔCt basal conditions/ΔCt test conditions. Results are shown as mean ± s.e.m. The data underlying this figure can be found at https://osf.io/wfgvs/.
(JPG)

**S5 Fig. HSF-1 is a minor player in the modulation of the stress response by INS-3. (A)** Survival percentages of wild-type animals*, ins-3* single mutants*, hsf-1* single mutants *and ins-3;hsf-1* double mutants exposed to oxidative (left) and heat stress (right). Results are shown as mean ± s.e.m. Five and six independent experiments were performed for oxidative stress and heat, respectively. Each experiment included 50–100 worms per condition. **(B)** Oxidation resistance of animals subjected to intestinal RNAi-mediated silencing of *hsf-1*. The VP303 (*kbIs7 [nhx-2p::rde-1 + rol-6(su1006)]*) background allows RNAi silencing only in the intestine [55,109,110]. RNAi, RNA interference. Results are shown as mean ± s.e.m. Results are shown as mean ± s.e.m. Four independent experiments were performed ($n = 4$). Each experiment included 30–80 worms per condition. The data underlying this figure can be found at https://osf.io/wfgvs/.
(JPG)

**S6 Fig. *ins-4* expression is enhanced upon oxidative stress.** Left. Representative epifluorescence images (20×) of young non-gravid adults expressing *Pins-4::GFP* under basal conditions, oxidative (2 h, 1 mM FeSO$_4$) or heat stress (6 h, 30 °C). Scale bar, 50 μm. Right. Corresponding quantification of fluorescence levels per worm. Scatter dot plot with relative expression of *Pins-4::GFP* normalized to the basal condition of each independent experiment. Line at the median. $n = 45$–60 animals per condition distributed across (three) independent experiments. One-way ANOVA (Kruskal–Wallis test) and Dunn's post hoc test versus basal were used. ns, not significant, ***$p < 0.001$. The data underlying this figure can be found at https://osf.io/wfgvs/.
(JPG)

**S7 Fig. Tyramine signaling mutants have no obvious defects in the defecation motor program. (A)** Representative kymographs of intestinal GCaMP fluorescence in freely behaving animals on NGM plates seeded with OP50. The anterior or the worm is oriented toward the top of each kymograph. The mean fluorescent intensity (white trace) is superimposed on the kymograph with a *y*-axis range of −1,000–8,000 arbitrary units (a.u.). Colormap values are indicated on the right. **(B)** Quantification of defecation interval during 5-min recordings in wild-type ($n = 7$ animals), *tdc-1* ($n = 6$ animals), and *tyra-3* ($n = 10$ animals) mutants. Defecation events occur every 45–50 s in wild-type, *tdc-1*, and *tyra-3* mutant animals. Results are mean ± s.e.m, one-way ANOVA with Dunnett's multiple comparison test. ns, not significant. The data underlying this figure can be found at https://osf.io/wfgvs/.
(JPG)

**S8 Fig. Exogenous tyramine alters intestinal Ca$^{2+}$ waves in tyramine receptor quadruple mutants.** Measurement of intestinal GCaMP fluorescence in *lgc-55; ser-2, tyra-2, tyra-3* quadruple mutants exposed to 30 mM tyramine. Kymographs are oriented with the anterior of the animal at the top. The mean fluorescent intensity (white trace) is superimposed on the kymograph with a *y*-axis range of −1,000–8,000 arbitrary units (a.u.). Colormap values for the kymograph are indicated on the right. Twenty-two animals were recorded in total. The data underlying this figure can be found at https://osf.io/wfgvs/.
(JPG)

**S9 Fig. INS-3 is tonically released from the intestine under basal conditions.** Representative image (40×) of L4-staged worm carrying P*ges-1*::*INS-3*::*VENUS* in basal conditions showed on differential interference contrast (DIC), fluorescence, and merged. Note the fluorescence both in the intestine (faint) and in the coelomocytes. Scale bar, 50 μm.
(JPG)

**S10 Fig. Exogenous tyramine does not up-regulate the transcription of *Pges-1*::*INS-3*:: VENUS.** Log$_2$ fold-changes in Venus transcript levels in transgenic animals expressing *Pges-1::ins-3::venus* exposed to exogenous tyramine (15 mM). Negative and positive values indicate down- and up-regulation of the transcript compared to non-exposed to non-exposed animals. Fold change was calculated as $\Delta Ct_{no\ Tyr}/\Delta Ct_{Tyr}$. Results are shown as mean ± s.e.m. The data underlying this figure can be found at https://osf.io/wfgvs/.
(JPG)

**S11 Fig. Tyraminergic modulation of stress resistance through intestinal INS-3 release requires the G-protein-coupled receptor TYRA-3.** Survival percentages to oxidative (left) and heat stress (right) of wild-type, *tyra-3* null mutants, and animals expressing the intestinal rescue of *ins-3* on wild-type and *tyra-3* null mutant backgrounds in the absence or presence of exogenous tyramine (15 mM), (mean ± s.e.m). Four independent experiments were performed (*n* = 4). Each experiment included 40–80 animals per condition. For conditions with tyramine, a two-tailed Student's *t* test versus the same strain without tyramine was used. ns, not significant, * *p* < 0.05. Two-tailed Student's *t* test was used between *tyra-3* null mutants and *tyra-3*; *ins-3*; P*elt-2*::*INS-3* strain. ns, not significant. The data underlying this figure can be found at https://osf.io/wfgvs/.
(JPG)

**S1 Video. Rhythmic Ca$^{2+}$ waves propagate from the posterior to the anterior end of the intestine in wild-type animals every 45–50 s.** Representative video showing intestinal calcium transients in a wild-type animal expressing GCaMP6 in the intestine in the presence of food.
(MP4)

**S2 Video. Exogenous tyramine (30 mM) triggers a dramatic increase in Ca$^{2+}$ transients in the intestine.** Representative video showing intestinal calcium transients generated by 30 mM tyramine in a wild-type animal expressing GCaMP6 in the intestine in the presence of food.
(MP4)

**S3 Video. Exogenous tyramine (30 mM) triggers high-frequency Ca2$^{+}$ transients even in the absence of food.** Representative video showing intestinal calcium transients generated by 30 mM tyramine in a wild-type animal expressing GCaMP6 in the intestine in the absence of food.
(MP4)

**S4 Video. Tyramine-triggered intestinal Ca2$^{+}$ transients are significantly reduced in *tyra-3* null mutants.** Representative video showing intestinal calcium transients generated by 30 mM tyramine in a *tyra-3* null mutant animal expressing GCaMP6 in the intestine.
(MP4)

**S5 Video. Tyramine-triggered intestinal Ca2+ transients are mostly abolished in *itr-1* null mutants.** Representative video showing intestinal calcium transients generated by 30 mM tyramine in an *itr-1* null mutant animal expressing GCaMP6 in the intestine.
(MP4)

**S1 Table. *ins-3* is expressed in *ins-3* null mutants rescued with the *Prgef-1::ins-3* transgene.** ΔCT values of *ins-3* measured by qPCR in *ins-3* null mutants and animals rescued with the *Prgef-1::ins-3* transgene. These data support effective neuronal expression of *ins-3* in the rescued animals.
(PDF)

## Acknowledgments

Some strains were provided by the CGC, which is funded by the NIH Office of Research Infrastructure Programs (P40 OD010440). We thank Andrés Garelli for the helpful discussions. In addition, we would like to acknowledge Ignacio Bergé, Andrea Thackeray, Adrian Bizet, Carolina Gomila, Marta Stulhdreher, and Carla Chrestía for technical support. We thank Marian Walhout for strains.

## Author contributions

**Conceptualization:** Tania Veuthey, Jeremy T Florman, Stefano Romussi, María José De Rosa, Mark J Alkema, Diego Rayes.

**Data curation:** Tania Veuthey, Jeremy T Florman, Sebastián Giunti, Stefano Romussi, Diego Rayes.

**Formal analysis:** Tania Veuthey, Jeremy T Florman, Sebastián Giunti, María José De Rosa, Diego Rayes.

**Funding acquisition:** Tania Veuthey, María José De Rosa, Mark J Alkema, Diego Rayes.

**Investigation:** Tania Veuthey, Jeremy T Florman, Sebastián Giunti, Stefano Romussi, María José De Rosa, Diego Rayes.

**Methodology:** Tania Veuthey, Jeremy T Florman, Sebastián Giunti, Stefano Romussi, Diego Rayes.

**Supervision:** María José De Rosa, Mark J Alkema, Diego Rayes.

**Validation:** Jeremy T Florman, Mark J Alkema, Diego Rayes.

**Writing – original draft:** Tania Veuthey, Mark J Alkema, Diego Rayes.

**Writing – review & editing:** Tania Veuthey, Jeremy T Florman, María José De Rosa, Mark J Alkema, Diego Rayes.

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
