## [Editor Report · Decision Letter 0]

22 Feb 2024

Dear Dr Rayes, 

Thank you for submitting your manuscript entitled "The neurohormone tyramine stimulates the secretion of an Insulin-Like Peptide from the intestine to modulate the systemic stress response in C. elegans" for consideration as a Research Article by PLOS Biology, and please accept our apologies for the delay in sending you an initial decision. We had wished to discuss your paper with an Academic Editor with relevant expertise, but had a bit of trouble finding someone who was available to provide advice over the last week. 

In the absence of input from an Academic Editor, I have discussed your study with my colleagues within the editorial team and I am writing to let you know that we would like to send your submission out for external peer review. I should note that, while we are in principle interested in the study, we do not yet feel able to make a firm call about whether the novel mechanistic insights provided here offer the level of conceptual advance needed for publication at PLOS Biology - and so we will be looking for strong reviewer support in that regard. 

Please note that before we can send your manuscript to reviewers, we need you to complete your submission by providing the metadata that is required for full assessment. To this end, please login to Editorial Manager where you will find the paper in the 'Submissions Needing Revisions' folder on your homepage. Please click 'Revise Submission' from the Action Links and complete all additional questions in the submission questionnaire.

Once your full submission is complete, your paper will undergo a series of checks in preparation for peer review. After your manuscript has passed the checks it will be sent out for review. To provide the metadata for your submission, please Login to Editorial Manager (https://www.editorialmanager.com/pbiology ) within two working days, i.e. by Feb 26 2024 11:59PM.

Kind regards,

Lucas

Lucas Smith, Ph.D.

Senior Editor

PLOS Biology

lsmith@plos.org

---

## [Decision Letter · Decision Letter 1]

5 Apr 2024

Dear Dr Rayes,

Thank you for your patience while your manuscript "The neurohormone tyramine stimulates the secretion of an Insulin-Like Peptide from the intestine to modulate the systemic stress response in C. elegans" was peer-reviewed at PLOS Biology. It has now been evaluated by the PLOS Biology editors, an Academic Editor with relevant expertise, and by several independent reviewers. 

In light of the reviews, which you will find at the end of this email, we would like to invite you to revise the work to thoroughly address the reviewers' reports.

As you will see in their comments below, the reviewers have raised a number of important concerns with the strength of conclusions and the novelty of the findings reported here - and we think these concerns would need to be thoroughly addressed before we could consider your study for publication. As noted in my previous correspondence, when sending your paper out for review, we were somewhat on the fence, editorially, about whether the study offered a sufficient conceptual advance for consideration at PLOS Biology, without providing deeper mechanistic insights, etc. 

Considering this, and after cross-reviewer discussion and after consulting with the Academic Editor, we think the study would need to be substantially expanded to be a strong candidate for further consideration at the journal. We do appreciate that the reviewers have suggested ways in which the study could be developed further, and so we would be willing to consider a revised version of your manuscript, if you are willing to experimentally address the reviewer comments and expand the study. For example, the reviewers have suggested that you should characterize if other stresses induce similar responses, elucidate the role of other tyramine receptors in this pathway, expand the mechanistic characterizations, and explore the implications of tyramine mutations on lifespan. 

We do understand that these requests represent a lot of work, with an uncertain outcome, and I must stress that we would only move forward with the study if the conceptual advance was expanded and the reviewers were supportive. We would therefore understand if you prefer to pursue faster publication of the study elsewhere. To that end, if you would like, we would be happy to discuss the possibility transferring your study to PLOS Genetics, perhaps with a more modest revision. If you would like to explore such a transfer, please let me know. (Note: the PLOS journals are editorially independent, and I cannot make guarantees at this stage about the outcome of the transfer discussion). 

We expect to receive your revised manuscript within 3 months. However, if you need additional time to complete the revision, we would be happy to grant an extension. Please email us (plosbiology@plos.org) if you have any questions or concerns, or would like to request an extension. 

Given the extent of revision needed, we cannot make a decision about publication until we have seen the revised manuscript and your response to the reviewers' comments. Your revised manuscript is likely to be sent for further evaluation by all or a subset of the reviewers.

**IMPORTANT - SUBMITTING YOUR REVISION**

*Re-submission Checklist*

*Published Peer Review*

*PLOS Data Policy*

Please note that as a condition of publication PLOS' data policy (http://journals.plos.org/plosbiology/s/data-availability ) requires that you make available all data used to draw the conclusions arrived at in your manuscript. If you have not already done so, you must include any data used in your manuscript either in appropriate repositories, within the body of the manuscript, or as supporting information (N.B. this includes any numerical values that were used to generate graphs, histograms etc.). For an example see here: http://www.plosbiology.org/article/info%3Adoi%2F10.1371%2Fjournal.pbio.1001908#s5

*Blot and Gel Data Policy*

Sincerely,

Lucas

Lucas Smith, Ph.D.

Senior Editor

PLOS Biology

lsmith@plos.org

REVIEWS:

Reviewer #1: Reviewer summary: 

Veuthey et al. provides evidence to demonstrate the role of INS-3 insulin-like peptide in modulating stress response in C. elegans. They also demonstrated that INS-3 release is induced by tyramine and acts via DAF-2 to modulate the stress response. given the group's prior Nature paper, some more novelty is needed. 

The following are some issue that needs to be addressed. 

Major:

1. Why only oxidative and thermal stress was chosen for the study? What about starvation stress (food-searching behavior or rate of motility) and introduction to pathogenic bacterial strains (percentage survival or reduced lifespan)? Does this pathway play any role in other environmental stress? It would be nice to have a supplementary figure showing the resistance (or behavioral change) of ins-3 mutant worms in other stress conditions such as acute starvation (food-searching behavior or rate of motility), lifespan assay with pathogenic bacterial strain (Pseudomonas aeruginosa), etc. 

2. Lines 133 - 137: The two sentences need more explanation. Heat stress for 30 min. in WT and heat stress for 10 min. in ins-3 mutants have the same outcome, i.e. nuclear accumulation of DAF-16. This isn't very clear and needs more explanation. The question that arises are 1) Why DAF-16 translocate to the nucleus in the WT condition? I would expect WT to respond differently compared to ins-3 mutant worms. 2) Why does DAF-16 take more time to move to the nucleus in WT worms? Is it because the initial flight response was converted into a cytoprotective response after prolonged exposure of 30 min.? If so, explain this. 3) Is there any other data that supports this explanation? If so, please include this data and explain appropriately. 

3. Is the fluorescence intensity quantification experiment (Fig. 3B, 4, and 5) repeated 3 times independently? If not, at least repeat one more time to confirm the reproducibility.

4. Figure 4: Representative worm images are not so convincing. Instead of having one worm in the field have many worms as shown below. If possible, apply this strategy to other images too. 

5. Figure 5: The authors showed tyramine treatment increased ins-3::venus fluorescence intensity in coelomocytes without additional transcription of Pges-1::ins-3::venus. It will be nice, as well as recommended, to show that the fluorescence intensity in intestinal tissue decreases in response to the release of INS-3. Also, can the authors use tyra-3 mutant worms to show that ins-3::venus did not increase in the coelomocytes? 

6. The model diagram (Figure 7B) can be improved by 1) adding a "?" to the unknown regions of the pathway (activation of TYRA-3 releasing the INS-3 vesicles), 2) Mark the signaling pathway that was previously found by citing the articles and 3) Mark the signaling pathway that is explained in this article.

7. What is the role of other tyramine receptors, such as ser-2, tyra-2, etc., in this pathway? It will be nice to see how mutants of other tyramine receptors behave under oxidative and thermal stress, at least as a supplemental figure. 

Minor:

Line 71 - Can you please mention the specific pair of neurons that release tyramine in this sentence? It will be helpful to know it right-a-way instead of going back to the citation. 

Line 119 - " ……promoter in a ins-3 null….." or "……..an ins-3 null………."?

Figure legend: When you mention "n=3", I understand that it is 3 experiments trials or biological replicates. It can be made clear by "n=3 experimental trials" or "n=40-50 worms per condition (in other figures)". For example: The authors have done this in figure 6A legend, "n=6-15 independent experiments". 

Figure 2A - Please mark the anterior-posterior directions of the worm and also use a dashed white line to outline the worm (similar to S4). Also, do this for all the figures. 

Figure 2 legend: Which graph uses a t-test? Please mention which graphs are analyzed using one-way ANOVA and which using t-test. Please follow this for all the figure legends. 

Figure 3A legend: "normalized to naïve animals", it is not completely understandable. What are naïve animals? If a normalization is done (heat shock animal/naïve animals) what is the proper label for the Y-axis in Fig 3A? Also, explain naïve animals in other figure legends. 

In the discussion, please discuss the role of other tyramine-producing cells such as uv1 cells near the vulva and gonadal sheath cells in stress response, if any?

Line 410 - Please add a space between 35 and mm in "35mm plates". As well as follow a uniform writing style instead of "35-mm" (line 423, 426, etc.). 

Line 414 - how many time the bacterial culture was concentrated? 

Line 450 - please write a few lines about the plugin used for fluorescence intensity quantification. 

Line 478 - please explain naïve condition. 

Line 494 - "elt-2" should be in italics. 

Line 525 - space between "30" and "mM" in "30mM".

General:

Check for "full stop" or "period" at the end of the sentences. It is missing in several places and make everything uniform throughout the manuscript. 

Please check the typos and punctuation throughout the manuscript. In some places, uniformity is missing, like "Scale bar, 50 µm" and in other places "Scale bar 50 µm", "Discussion", etc. 

Reviewer #2: In this study by Veuthey et al., the authors highlight that a mutant form of the insulin-like peptide ins-3, acting as an agonist for the DAf-2 pathway, displays incresed resistance to heat and oxidative stress. Their research underscores the significance of tyramine secretion via tyraminergic neurons in facilitating the systemic impact of ins-3 release from the intestine, ultimately enhancing stress resilience by activating DAF-16 within the intestine of C. elegans.

This study builds upon their previous work, where they initially observed that neural stress hormones trigger the release of ILPs from the intestine, influencing stress responses. Their current focus on the ILP INS-3 reveals that reduced ins-3 levels or expression amplify DAF-16 activity in the intestine.

The major issue of this work is their overall conclusion, that tyramine-induced ins-3 release in the intestine leads to heat stress resistance via DAF-16 and HSF-1 activation, which is not supported by experimental evidence. In particular, the sequence of events as proposed in their model in Figure 7 lacks experimental evidence. While there is some evidence for DAF-16 to be involved in this response, there is no data at all regarding HSF-1. Another issue is that the study heavily relies on fluorescence intensity measurements and reporter strains, which are not sufficiently conclusive. 

Specific points: 

Figure 3B and Figure 4: Confirm expression levels by Western blot using GFP antibody. A major shortcoming of this study is the lack of biochemical evidence by measuring protein expression using Western Blot analysis. Simply quantifying fluorescence intensity of hsp-16p::GFP and ins-3p::GFP is not sufficient here.

Figure 5: the enhanced fluorescence intensity of the ges-1p::ins-3::Venus reporter is not a consequence of tyramine inducing an increase in ins-3 transcription. The ges-1p promoter is an intestine-specific promoter that overexpresses ins-3 in the intestine: is not under the control of the ins-3 endogenous promoter. This means that the conclusion they are drawing from this result cannot be accurate. To really confirm that ins-3 transcripts are not induced by tyramine, qPCR measurements of ins-3 mRNA is necessary. Moreover, it is unclear why the expression of ins-3::venus under control of ges-1p (an intestine-specific promoter) does not show any fluorescence in the intestine. 

Figure 6B: Figure 6B's findings that rescuing ins-3 in tyra-3 and tdc-1 mutants doesn't significantly impact stress resistance suggests that tyraminergic signaling likely has other targets contributing to stress resilience and that ins-3 is not the sole contributor that modulates stress survival. If it were, then the expectation is that the intestinal ins-3 rescue strain suppresses the increased stress resistance of the tyra-3,ins-3 and tdc-1,ins-3 double mutnats. Moreover, the correct statistical analysis comparing tdc-1,ins-3 and tyra-3,ins-3 double mutants with the rescued intestinal ins-3 to the same genetic background is also necessary here. 

Figure 7: The continued conclusion throughout the paper that the increased release of INS-3 inhibits DAF-16 and HSF-1 activity is unfounded. It is necessary to show that INS-3 acts through the DAF-2 signaling pathway in the intestine. One way to show this is by intestine specific knockdown of HSF-1 and DAF-16. Does intestine-specific knockdown of hsf-1 and daf-16 in the genetic background of ins-3, tdc-1 or tyra-3 mutants still lead to increased stress resistance? How does this alter expression of DAF-16 and HSF-1 targets? It will be necessary to measure transcripts or provide Western blot analysis of HSF-1 targets such as hsp-70. hsp-16 is not a proxy for HSF1 activity as it is under control of both DAF-16 and HSF-1. 

Moreover, the study's implication of the daf-2/ILS pathway's raises questions into potential effects of tyramine mutants on lifespan, a direction that could provide deeper insights into the mechanisms at play.

Reviewer #3: Veythey et al manuscript on neurohormone tyramine that stimulates insulin-like peptide from intestine to modulate systemic stress in C. elegans

Strengths of this paper is it adds to the studies that investigate the role insulin-like peptides have in stress responses and the signaling pathway from gut and signaling pathway; C. elegans is a good system to investigate this.

The following need to be adressed (most minor issues)

1. Methodology: The heatshock protocol involved putting plates, sealed with parafilm, placed in a ziplock bag, and immersed in waterbath for 20hrs. Authors likely did this to ensure uniform temperature exposure to the worms. My concern is that the O2 levels could fluctuate in this environment. Unsure if they have thought of hypoxia as a compounding issue. 

Also within methods, the strain list (line 379)- may want to add references for each strain that was developed elsewhere.

Results:

2. Line 443 period at end of sentence missing and paragraph return typo? Line 456 period at end of sentence missing (check for typos throughout the manuscript)

3. Figure 2A state the larval stage of the animal 

4. Figure 2B - was RTPCR done to ensure that the PrGEF-1 (neurons) did indeed induce expression of ins-3. That is, the non-rescue isn't because of the ins-3 not being properly expressed in the neurons but that the neuronal ins-3 does not have a role in the survival.

5. Line 145/146 - add a reference that shows hsp-16.2 is an HSF-1 effector gene and refer to hsp-16.2 as such (instead of hsp-16).

6. Fig S3- it does help having the reporter noted in the figure (like Fig S1) and not just legend so perhaps add "ins-4 expression" to top of figure

7. Fig 4 and sentence on line 175-177- it looks like ins-3::GFP is expressed in the posterior part of the intestine when exposed to exogenous tyramine. Add "posterior" to further specify (levels do not seem throughout the intestine). Has higher magnification of specific areas of the intestine been analyzed to know which intestinal cells have the expression of ins-3::GFP in response to tyramine.

8. In the text that states "young adult" can the age be clarified and if gravid or not (1 day old vs non gravid after molting); just for clarity.

9. Tyramine triggering INS-3 release - why wasn't INS-3 translational reporter with endogenous promotor (e.g. CRISPR strain) used; The expression level of Pges-1::ins-3::venus seems pretty low given it is driven by Pges-1. Also can authors clarify if the venus marker was used to minimize autofluorescent issue from the intestine and that autofluorescence not observed.

---

## [Decision Letter · Decision Letter 2]

4 Dec 2024

Dear Dr Rayes,

Thank you for your patience while we considered your revised manuscript "The neurohormone tyramine stimulates the secretion of an Insulin-Like Peptide from the intestine to modulate the systemic stress response in C. elegans" for publication as a Research Article at PLOS Biology. Your revised study has been evaluated by the PLOS Biology editors, the Academic Editor and the original reviewers.

The reviews are appended below. As you will see the reviewers agree that the study has been strengthened in the last revision, and Reviewers 1 and 3 are now fully satisfied and suggest that we accept the study. However, Reviewer 2 has a few lingering concerns and we think that these should be addressed before publication. Specifically, reviewer 2 has commented that more evidence is needed to show that the hsf1 mutant is not sensitive to heat shock, and maintains that it would be important to corroborate GFP fluorescence experiments with western blot data. After discussing these points with the Academic Editor, we think these last experiments would be important to fully support the conclusions of the study and so encourage you to add this data. 

We would therefore like to invite you to address the last reviewer points in another revision. While we think the scope of the remaining revisions needed is relatively straightforward, given that some new data is needed and given the upcoming holidays, we have provided a 3 month deadline for the next revision. Please email us (plosbiology@plos.org) if you have any questions or concerns, or if you end up needing an extension.

Given the extent of revision needed, we cannot make a decision about publication until we have seen the revised manuscript and your response to the reviewers' comments. Your revised manuscript may be sent out for further evaluation by a subset of the reviewers.

**IMPORTANT - SUBMITTING YOUR REVISION**

*Re-submission Checklist*

*Published Peer Review*

*PLOS Data Policy*

Please note that as a condition of publication PLOS' data policy (http://journals.plos.org/plosbiology/s/data-availability ) requires that you make available all data used to draw the conclusions arrived at in your manuscript. If you have not already done so, you must include any data used in your manuscript either in appropriate repositories, within the body of the manuscript, or as supporting information (N.B. this includes any numerical values that were used to generate graphs, histograms etc.). For an example see here: http://www.plosbiology.org/article/info%3Adoi%2F10.1371%2Fjournal.pbio.1001908#s5

*Blot and Gel Data Policy*

Sincerely,

Lucas

Lucas Smith, Ph.D.

Senior Editor

PLOS Biology

lsmith@plos.org

REVIEWS:

Reviewer #1: I am happy with the author's response and recommend the manuscript for publication. 

Reviewer #2: While the authors have addressed all previous comments and the manuscript has significantly improved, I still have some concerns.

Firstly, although the hsf-1(sy441) mutant has been demonstrated to be sensitive to heat shock in numerous prior studies, the heat-shock conditions used here (exposing C. elegans to 35°C for only 4 hours) may not be sufficient to observe this sensitivity. If the authors wish to claim that the hsf-1 mutant is not sensitive to heat shock and that HSF-1 plays only a minor role in the modulation of stress responses by INS-3, I recommend conducting a heat-shock experiment using hsf-1 RNAi (knockdown at the L2 stage). Another issue with this is that there has been a lot of controversy with regards to the hsf-1(sy441) mutant. Some labs observed the reduced thermotolerance of this mutant during heat shock conditions, whereas others observed increased heat stress resistance that is age-dependent (Morton et al, 2013; Golden et al, 2020, and Kovacs et al., Aging Cell 2024). This should be considered in the authors response and revised manuscript.

Secondly, I disagree with the assertion that measuring GFP fluorescence intensity as a proxy for protein expression in C. elegans. While measuring GFP fluorescence intensity is indeed accepted in the C. elegans field, it is not as rigorous and accurate as Western blot analysis. Indeed, many research papers utilize Western blotting with specific antibodies to quantify protein expression also in C. elegans (e.g. Tian et al., Cell 2016, Charmpilas et al, Cell Reports 2024). 

Reviewer #3: Thank you for addressing reviewer comments- interesting article.

---

## [Editor Report · Decision Letter 3]

20 Dec 2024

Dear Dr Rayes,

Thank you for the submission of your revised Research Article "The neurohormone tyramine stimulates the secretion of an Insulin-Like Peptide from the intestine to modulate the systemic stress response in C. elegans" for publication in PLOS Biology. I have now discussed your revision with the Academic Editor, Ursula Jakob, and we are satisfied by the changes made and convinced by your rebuttals. Therefore, on behalf of my colleagues and the Academic Editor, I am pleased to say that we can in principle accept your manuscript for publication, provided you address any remaining formatting and reporting issues. These will be detailed in an email you should receive within 2-3 business days from our colleagues in the journal operations team; no action is required from you until then. Please note that we will not be able to formally accept your manuscript and schedule it for publication until you have completed any requested changes.

**IMPORTANT: As you address any formatting and reporting requests, to come, please also attend to the following editorial requests as well: 

1) TITLE: We would like to suggest a tweak to your title, to make it a bit more grammatically tight. If you agree, we suggest you change it to: 

"The neurohormone tyramine stimulates the secretion of an Insulin-Like Peptide from the C. elegans intestine to modulate the systemic stress response"

2) DATA: thank you for providing the underlying data for your study on OSF. Can you please update each relevant figure legend with a sentence directing readers to this dataset? For example, to each figure legend you can add the sentence "the data underlying this figure can be found at https://osf.io/wfgvs/

3) CODE: Thanks also for including the code for the macros used in your study on OSF. Can you please update your data availability statement, in our editorial manager system to indicate this? For example, you can update the data statement to read "All the raw data and code are available on the Open Science Framework platform (https://osf.io/wfgvs/)"

PRESS

We also ask that you take this opportunity to read our Embargo Policy regarding the discussion, promotion and media coverage of work that is yet to be published by PLOS. As your manuscript is not yet published, it is bound by the conditions of our Embargo Policy. Please be aware that this policy is in place both to ensure that any press coverage of your article is fully substantiated and to provide a direct link between such coverage and the published work. For full details of our Embargo Policy, please visit http://www.plos.org/about/media-inquiries/embargo-policy/ .

Sincerely, 

Lucas Smith, Ph.D.

Senior Editor

PLOS Biology

lsmith@plos.org